# Using adopted individuals to partition indirect maternal genetic effects into prenatal and postnatal effects on offspring phenotypes

**Liang-Dar Hwang[1], Gunn-Helen Moen[1,2,3,4,5], David M Evans[1,2,6]***

[1]Institute for Molecular Bioscience, The University of Queensland, Brisbane, Australia; [2]The University of Queensland Diamantina Institute, The University of Queensland, Brisbane, Australia; [3]Institute for Clinical Medicine, Faculty of Medicine, University of Oslo, Oslo, Norway; [4]K.G. Jebsen Center for Genetic Epidemiology, Department of Public Health and Nursing, Norwegian University of Science and Technology, Trondheim, Norway; [5]Population Health Science, Bristol Medical School, University of Bristol, Bristol, United Kingdom; [6]MRC Integrative Epidemiology Unit, University of Bristol, Bristol, United Kingdom

**\*For correspondence:**
d.evans1@uq.edu.au

**Competing interest:** The authors declare that no competing interests exist.

**Abstract** Maternal genetic effects can be defined as the effect of a mother's genotype on the phenotype of her offspring, independent of the offspring's genotype. Maternal genetic effects can act via the intrauterine environment during pregnancy and/or via the postnatal environment. In this manuscript, we present a simple extension to the basic adoption design that uses structural equation modelling (SEM) to partition maternal genetic effects into prenatal and postnatal effects. We examine the power, utility and type I error rate of our model using simulations and asymptotic power calculations. We apply our model to polygenic scores of educational attainment and birth weight associated variants, in up to 5,178 adopted singletons, 943 trios, 2687 mother-offspring pairs, 712 father-offspring pairs and 347,980 singletons from the UK Biobank. Our results show the expected pattern of maternal genetic effects on offspring birth weight, but unexpectedly large prenatal maternal genetic effects on offspring educational attainment. Sensitivity and simulation analyses suggest this result may be at least partially due to adopted individuals in the UK Biobank being raised by their biological relatives. We show that accurate modelling of these sorts of cryptic relationships is sufficient to bring type I error rate under control and produce asymptotically unbiased estimates of prenatal and postnatal maternal genetic effects. We conclude that there would be considerable value in following up adopted individuals in the UK Biobank to determine whether they were raised by their biological relatives, and if so, to precisely ascertain the nature of these relationships. These adopted individuals could then be incorporated into informative statistical genetics models like the one described in our manuscript to further elucidate the genetic architecture of complex traits and diseases.

## Editor's evaluation

The authors propose an innovative and sound method to leverage the adoption of a design for disentangling prenatal and postnatal genetic effects. This work will be of interest to scientists interested in the intergenerational transmission of phenotypes through genetic pathways.

## Introduction

Maternal genetic effects can be defined as the causal influence of maternal genotypes on offspring phenotypes over and above that which results from the transmission of genes from mothers to their offspring (*Wolf and Wade, 2009*). Over the last few years there has been a resurgence of interest in identifying and quantifying maternal genetic effects on offspring phenotypes, both from the perspective of variance component estimation (*Balbona et al., 2021*; *Bates et al., 2018*; *Eaves et al., 2014*; *Eilertsen et al., 2021*; *Kim et al., 2021*; *Kong et al., 2018*; *Qiao et al., 2020*; *Tubbs et al., 2020*) and estimating the causal effect of specific maternal environmental exposures on offspring outcomes through Mendelian randomization based approaches (*Evans et al., 2019*; *Lawlor et al., 2017*; *Moen et al., 2020*; *Tyrrell et al., 2016*; *Warrington et al., 2019*; *Zhang et al., 2015*). Indeed, following the advent of transgenerational genome-wide association studies, maternal genetic effects are beginning to be identified at individual genetic loci (*Warrington et al., 2019*; *Beaumont et al., 2018*), a trend that is set to continue as the sample sizes of such studies increase further. Given the increasing number of variants identified in GWAS that exhibit robust maternal genetic effects, a natural question to ask is whether these loci exert their effects on offspring phenotypes through intrauterine mechanisms, the postnatal environment, or both. Indeed, resolving maternal effects into prenatal and postnatal sources of variation could be a valuable first step in eventually elucidating the underlying mechanisms behind these associations (*Armstrong-Carter et al., 2020*), directing investigators to where they should focus their attention, and in the case of disease-related phenotypes, yielding potentially important information regarding the optimal timing of interventions. For example, the demonstration of maternal prenatal effects on offspring IQ/educational attainment, suggest that if the mediating factors that were responsible could be identified, then improvements in the prenatal care of mothers and their unborn babies targeting these factors, could yield useful increases in offspring IQ/educational attainment.

In this manuscript, we develop a simple method for partitioning maternal genetic effects into prenatal and postnatal components that leverages information provided by adopted individuals. We assume that an adopted individual's phenotype is influenced by prenatal intrauterine factors as proxied by their biological mother's genome, and postnatal influences as proxied by their adoptive mother's and father's genomes. In contrast, we assume that the phenotype of individuals who have not been adopted are influenced by prenatal intrauterine factors resulting from their biological mother's genome, and postnatal factors as proxied by their biological mother's and father's genomes. This model leads to different expectations for the covariance between an individual's phenotype and their own and their relatives' genotypes depending on whether they have been adopted or not.

One of the challenges in applying this kind of framework to real life situations is the paucity of cohorts containing large numbers of adoptive families (*Horn, 1983*; *Rhea et al., 2013*; *Scarr and Weinberg, 1983*). Restricted numbers of adoptive families will consequently limit the statistical power to partition maternal genetic effects- particularly at single genetic variants which tend to have very small effect sizes. It is important to realize, however, that adopted 'singletons' (i.e. adopted individuals whose adoptive and biological parents have not been included in the study) provide important information on partitioning maternal genetic effects into prenatal and postnatal contributions regardless of whether information on their parents has been gathered. The intuition behind this surprising fact is that the covariance between an adopted individual's genotype and phenotype is a function of prenatal (but not postnatal) maternal genetic effects (*Figure 1*, G5 to G7). In contrast, the covariance between a non-adopted individual and their own phenotype includes contributions from prenatal and postnatal genetic effects (*Figure 1*, G1 to G4). Thus, the difference between the genotype-phenotype covariance in adopted and non-adopted singleton individuals provides important information on the likely size of postnatal genetic effects. This is potentially important given that >7000 individuals in the UK Biobank study (*Sudlow et al., 2015*) report being adopted, and hence this publicly available dataset may represent a powerful resource for identifying prenatal maternal effects and partitioning maternal genetic effects into prenatal and postnatal sources of variation.

In this manuscript, we introduce a new structural equation model (SEM) to estimate prenatal and postnatal maternal genetic effects on offspring phenotypes. The SEM framework allows us to model the contributions from missing individuals as latent variables, facilitating the inclusion of information from adopted singleton individuals whose adoptive and/or biological parents are not available for analysis. We estimate the asymptotic power of our model to partition maternal genetic effects into

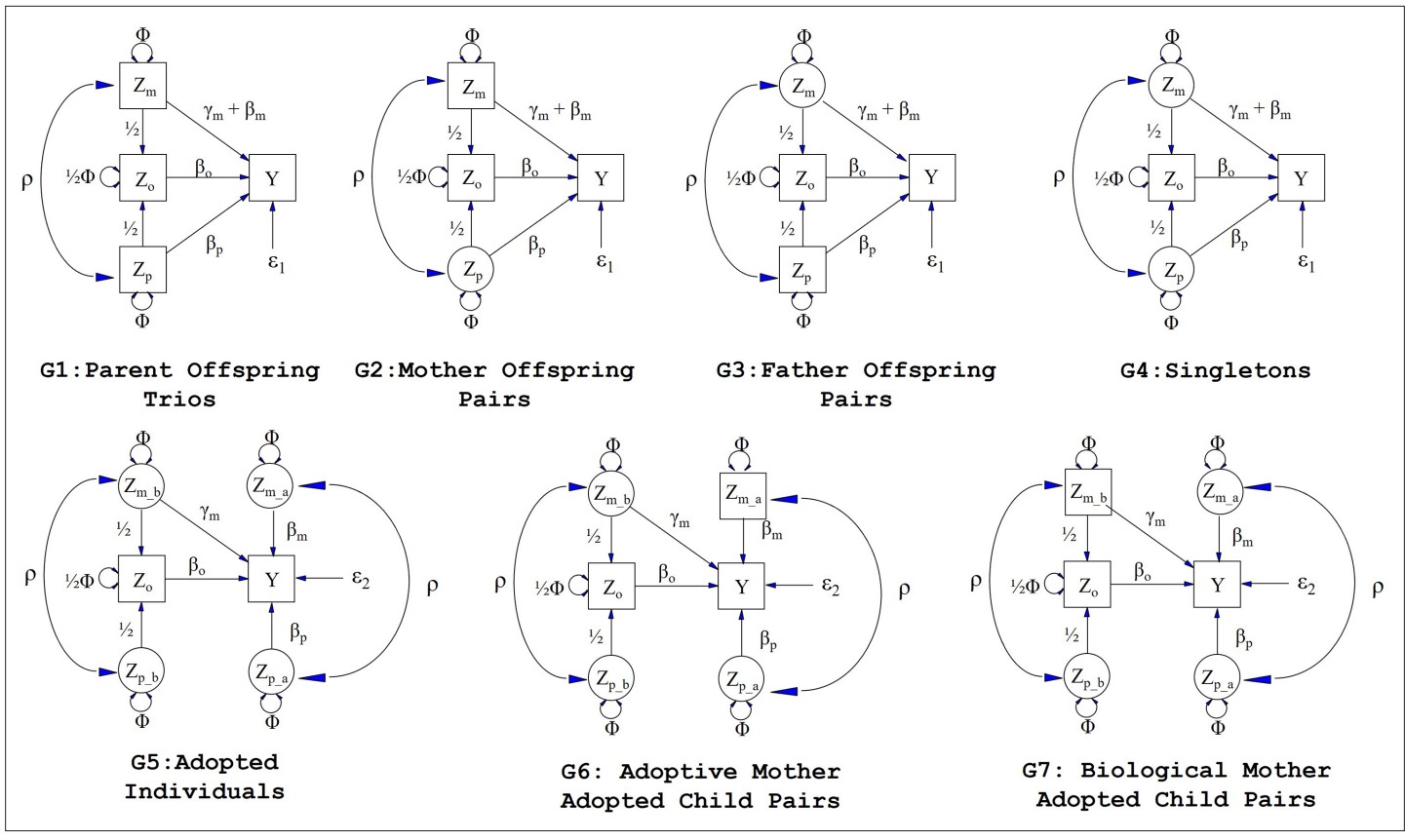

**Figure 1.** Path diagrams illustrating the structural equation models (SEM) underlying the seven family structures modelled in this manuscript (G1 – G7). Causal relationships are represented by one headed arrows. Two headed arrows represents correlational relationships. Observed variables and latent variables are shown in squares and circles, respectively. $Z_m$ represents biological mother's genotype which influences offspring phenotype (**Y**) via prenatal ($\gamma_m$) and postnatal ($\beta_m$) pathways. $Z_p$ represents the biological father's genotype which only influences offspring phenotype postnatally ($\beta_p$). $Z_o$ represents offspring genotype which influences offspring phenotype ($\beta_o$) and is correlated ½ with the genotypes of its biological parents. $Z_{m\_b}$ represents the genotype of a biological mother whose child was adopted and therefore only influences her child's phenotype through prenatal pathways ($\gamma_m$). $Z_{m\_a}$ represents the adoptive mother's genotype which only influences her adopted offspring's phenotype via postnatal pathways ($\beta_m$). $Z_{p\_b}$ represents the genotype of a biological father whose child was adopted and therefore has no influence on the adopted offspring phenotype. $Z_{p\_a}$ represents the adoptive father's genotype which influences his adopted offspring postnatally ($\beta_p$). $\rho$ represents the covariance between parental genotypes, as a result of e.g. assortative mating (it is assumed that this covariance is the same in biological parents and adoptive parents). The total variance of genotypes in the parental generation is set to $\Phi$. $\varepsilon_1$ and $\varepsilon_2$ represent residual error terms for the biological and adopted offspring phenotypes respectively that we assume have different variances.

prenatal and postnatal components and confirm these results by simulation. We code our routines into a freely available R shiny app web utility that researchers can use to perform their own power calculations. Finally, we apply our methods to birth weight and educational attainment data in the UK Biobank- two phenotypes known to be affected by maternal genetic effects. We investigate the effect of inadvertently including adopted singleton individuals whose adoptive parents are genetically related to them in our model, adjust our model to correct for this misspecification and quantify the effect of this adjustment on type 1 error rates and power. We discuss the implications of model misspecification for using the UK Biobank resource in this context in future studies more broadly.

## Materials and methods
### Model description

We consider seven different family structures (*Figure 1*) which we model using SEM (although we note that the SEM framework is flexible enough to include many other sorts of family structures as well). The different family structures we consider are: Biological parent-offspring trios (G1), biological mother-offspring pairs (G2), biological father-offspring pairs (G3), singleton individuals (who

were raised by their biological parents) (G4), adopted singleton individuals (who were raised by their adoptive parents) (G5), adoptive mother-adopted child pairs (G6), and biological mother-adopted child pairs (G7). For each of these different family structures, we model missing parental genotypes (whether biological or adoptive) as latent variables.

Let the variables $Z_m$, $Z_p$ and $Z_o$ represent maternal, paternal and offspring genotypes. Likewise, let $Z_{m\_b}$ and $Z_{p\_b}$ denote the genotypes of the biological mother and father of an adopted individual respectively, and let $Z_{m\_a}$ and $Z_{p\_a}$ represent the genotypes of the adoptive mother and father of the adopted individual. These variables may be observed (i.e. the square boxes in *Figure 1*) or unobserved latent variables (i.e. the circles in *Figure 1*). They may represent polygenic risk scores (PRS) consisting of multiple genetic variants, or single genetic variants if the study is of sufficient size to reliably demonstrate genetic effects at individual loci. The model includes path coefficient terms for maternal genetic effects ($\gamma_m$, $\delta_m$), paternal genetic effects ($\delta_p$) and offspring genetic effects ($\delta_o$) on the offspring phenotype (Y). We are specifically interested in partitioning maternal genetic effects into prenatal ($\gamma_m$) and postnatal genetic effects ($\delta_m$) and we assume that the effect of the prenatal maternal genetic effect and postnatal maternal genetic effect on the offspring phenotype is additive (i.e. $\gamma_m + \delta_m$). We assume in the case of biological families (i.e. G1 to G4 in *Figure 1*), that biological mothers exert prenatal and postnatal genetic effects on their offspring. In contrast, in the case of adopted individuals (i.e. G5 to G7 in *Figure 1*), we assume that prenatal maternal effects from the biological mother and postnatal maternal genetic effects from the adoptive mother contribute to variation in the offspring phenotype. We assume that biological fathers exert only postnatal genetic effects on their offspring (biological families only), and adoptive fathers exert only postnatal genetic effects on their adopted offspring.

We estimate the genotypic variance ($\Phi$) of each individual and assume that it is the same in mothers, fathers, their children, and across both biological and adopted individuals under random mating (although the requirement for equal variances across individuals can be relaxed if necessary). We account for possible covariance between maternal and paternal genotypes (e.g. through assortative mating) by estimating the covariance path $\rho$. This is equivalent to modelling one round of assortative mating in the parental generation and has the effect of inflating the total genotypic variance in the offspring generation from $\Phi$ under random mating to $\Phi + \frac{1}{2}\rho$. Finally, to accommodate the possibility that adopted individuals may have different phenotypic variances to individuals raised by their biological relatives, we permit the variance of residual sources of variation to differ across adopted and biological family structures (*Cheesman et al., 2020*).

We caution that for all the parameters in the model to be identified, different constellations of relatives must be sampled from G1-G7. For example, to partition maternal genetic effects into prenatal and postnatal components, information from adopted individuals must be available. In fact, even adopted 'singletons', for whom there is no genotype information from their parents (i.e. biological or adoptive parents), contribute important information for the partitioning of maternal genetic effects, since the covariance between their own genetic score and phenotype is a function of offspring genetic effects and prenatal maternal effects, but not postnatal maternal effects (*Figure 1*). This contrasts with the situation in non-adopted individuals whose genotype-phenotype covariance is a function of all three sources of variation (plus postnatal paternal genetic effects). Thus, the difference between the genotype-phenotype covariance in adopted and non-adopted individuals provides important information on the size of postnatal maternal genetic effects. Indeed, the inclusion of adopted singleton individuals in our model is sufficient for identification as long as there are at least some (biological) parent-offspring trios (or alternatively both biological mother-child and biological father-child pairs) available. Whilst the presence of adopted individuals' parents (biological or adoptive) is not a necessary condition for partitioning maternal effects into prenatal and postnatal components, the inclusion of such pairs is advantageous in terms of statistical power as we show below. We include an example R script that fits the sort of SEM described in this article that users can modify to apply to their own data (see Source Code File).

## Asymptotic power calculations

We used OpenMx (*Boker et al., 2011*; *Neale et al., 2016*) to calculate the asymptotic power to partition maternal genetic effects into prenatal and postnatal maternal genetic effects. Asymptotic covariance matrices were generated assuming certain underlying values for the parameters of the

model depicted in *Figure 1*. The full model was then fitted to the covariance matrices to confirm a perfect fit to the data and to check that the parameter values were correctly estimated. Secondly, a reduced model where the parameter(s) of interest was constrained to zero was fitted to the same covariance matrices. We examined the effect of constraining the prenatal maternal genetic effect or postnatal maternal genetic effect to zero. The difference in minus twice the log-likelihood chi-square between the full and reduced models is equal to the non-centrality parameter of the test for association, with the degrees of freedom equal to the difference in the number of free parameters between the models. Power was then calculated as the area under the curve of a non-central chi-square distribution to the right of the significance threshold of interest:

$$Power = \int_{X_\alpha'^2(v,0)}^{\infty} dX'^2 (v, \varsigma),$$

where $X_\alpha'^2 (v, 0)$ is the quantile of the 100 * (1-α) percentage point of the central $\chi^2$ distribution with $v$ degrees of freedom, and $\varsigma$ is the non-centrality parameter (*Moen et al., 2019*).

As we were interested in whether there might be enough adopted individuals in the UK Biobank for the informative partitioning of maternal genetic effects into prenatal and postnatal sources of variation, we first calculated power ($\alpha$=0.05) using sample sizes roughly similar to the number of European individuals in the resource who reported their educational attainment (i.e. 1000 parent-offspring trios, 4000 mother-offspring pairs, 1800 father-offspring pairs, 300,000 singletons who were raised by their biological parents, 6000 adopted individuals and 50 biological mother-adopted offspring duos- see below). We investigated the effect of different combinations of prenatal maternal genetic effects, postnatal maternal genetic effects, postnatal paternal genetic effects and offspring genetic effects (i.e. $\gamma_m$ or $\delta_m$ = 0, 0.05, 0.1, 0.3 and 0.5). We also examined the effect of modifying the covariance between parental genotypes (with $\rho$ ranging between 0 and 0.2 times the genotypic variance in the parental generation). Finally, we were curious as to the effect of increasing the number of the different family structures on statistical power, and in particular, on whether including increasing numbers of more attainable/accessible biological relatives (i.e. biological trios, pairs, and singletons [G1 – G4]) could increase power to partition maternal genetic effects for a fixed number of adopted individuals. We also investigated the effect of varying the sample size of adopted individuals (G5), adoptive mother-adopted offspring duos (G6), and biological mother-adopted offspring duos (G7) on power. We used our calculator to determine 80% power ($\alpha$=0.05).

## Data simulations

In order to confirm our asymptotic calculations, we simulated a series of datasets of the same size and with the same underlying parameters as in the asymptotic power calculations. Each condition was simulated 1000 times and power was calculated as the number of simulations in which the effect of interest was detected (p<0.05) divided by 1000.

## Online web utility

We developed a freely available, online power calculator that allows investigators to explore the power to detect prenatal and postnatal maternal genetic effects using our SEM in their own studies (https://evansgroup.di.uq.edu.au/ADOPTED). Our power calculator is built using the R shiny app (https://shiny.rstudio.com/) running the R studio software (*RStudioTeam, 2015*) and the OpenMx package (*Boker et al., 2011*; *Neale et al., 2016*) in the background.

## Application to UK biobank data

We applied our model to self-reported birth weight and educational attainment data from the UK Biobank resource. The UK Biobank study was approved by the UK National Health Service National Research Ethics Service. Written consent was obtained from both the participants and their parents (for subjects younger than 18 years old). This study was approved by the Human Research Ethics Committee at the University of Queensland (approval number: 2019002705). We only included individuals of European ancestry, determined by: (i) projecting their genetic principal components onto the 1,000 Genome sample and then K-means clustering or (ii) those who self-reported ethnic background of either 'British', 'Irish', 'Irish', 'White', or 'Any other white background'. Family relationships

were determined by the pairwise kinship estimated across the whole UK Biobank sample using the software KING (*Manichaikul et al., 2010*). Adoption status was determined based on response to the question "*Were you adopted as a child?*". One from each pair of Individuals showing genome-wide genetic similarity greater than third degree relatives was removed from the dataset. We chose exclusions so as to maximize the number of families from adopted singletons (G5), followed by trios (G1), mother-offspring duos (G2), and then father-offspring duos (G3).

Birth weight is a phenotype that is known to be affected by the maternal genome (*Warrington et al., 2019*), but should only be influenced by prenatal and not postnatal maternal genetic effects. We only included individuals whose birth weight was between 2.5 kg and 4.5 kg in analyses. These included 731 biological trios, 2014 biological mother-offspring pairs, 485 biological father-offspring pairs, 168,522 singletons raised by their biological parents, 1084 singleton individuals who reported being adopted, and 16 biological mother-adopted offspring pairs. We calculated each individual's weighted PRS (i.e. sums of the birth weight-increasing allele weighted by the effect size) and unweighted PRS (i.e. counts of the birth weight increasing allele) for birth weight using 20 SNPs known to influence birth weight via maternal pathways with $p < 5 \times 10^{-8}$ from a recent GWAS of birth weight (*Warrington et al., 2019*). See *Supplementary file 1* for a list of SNPs used to construct PRS of birth weight. We regressed birth weight on year of birth, sex and the first five genetic principal components and used the residuals in the analysis.

We then applied our model to educational attainment in UK Biobank participants of European ancestry. We calculated years of education, as a proxy of educational attainment, based on response to the question "*Which of the following qualifications do you have?*": College or university degree = 20 years; Vocational Qualification (NVQ), Higher National Diploma (HNC), or equivalent = 19 years; other professional qualification = 15 years; A level/AS level or equivalent = 13 years; O level/General Certificate of Secondary Education (GCSE) or equivalent = 10 years; Certificate of Secondary Education (CSE) or equivalent = 10 years; none of the above = 7 years of education. We regressed years of education on year of birth, sex and the first five genetic principal components and used the residuals in the analysis. The final dataset included 943 biological parent-offspring trios, 2687 biological mother-offspring pairs, 712 biological father-offspring pairs, 347,980 singletons from biological families, 5178 adopted singletons and 39 biological mother-adopted offspring pairs. We calculated weighted PRS (i.e. sums of the educational attainment increasing allele and weighted by the effect size) and unweighted PRS (i.e. counts of the educational attainment increasing allele) using 1267 SNPs from the current largest GWAS of educational attainment (*Lee et al., 2018*). We included only genome-wide significant SNPs in the construction of these scores because the most recent GWAS meta-analysis of educational attainment by *Lee et al., 2018* included UK Biobank individuals (*Lee et al., 2018*). We were concerned that sample overlap would inflate estimates of the association between the PRS and educational attainment (and any inflation would be most severe for the hundreds of thousands of SNPs that did not reach genome-wide significance) and in turn bias parameter estimates from our model. For this reason, we also performed a sensitivity analysis using a second set of PRS of educational attainment using 72 SNPs from a GWAS of non-UK Biobank individuals (*Okbay et al., 2016*). See *Supplementary file 2* for a list of SNPs used to construct these PRS.

The UK Biobank contains only very limited information regarding the adoption status of individuals (i.e. only whether they report being adopted or not). We were concerned about the possibility that some individuals in UK Biobank might have been adopted by their biological relatives in which case our model in *Figure 1* would be misspecified. We therefore also conducted a sensitivity analysis by excluding adopted individuals with known breastfeeding information (on the reasoning that these individuals might be more likely to be fostered by a biologically related individual) and repeating the educational attainment analyses. This reduced the number of adopted 'singleton' individuals in the analysis to 2,867.

## Investigating the effect of model misspecification

We were concerned about the effect that misclassifying adopted "singleton" individuals as unrelated to their adoptive parents (i.e. when in reality they were raised by their biological relatives) might have on type I error rates and estimates of prenatal and postnatal maternal genetic effects generated by our model. We therefore simulated data under four scenarios: (i) adopted singletons where 0%, 20%, 40%, 60%, or 80% had adoptive mothers who were siblings or cousins of their biological mothers, (ii)

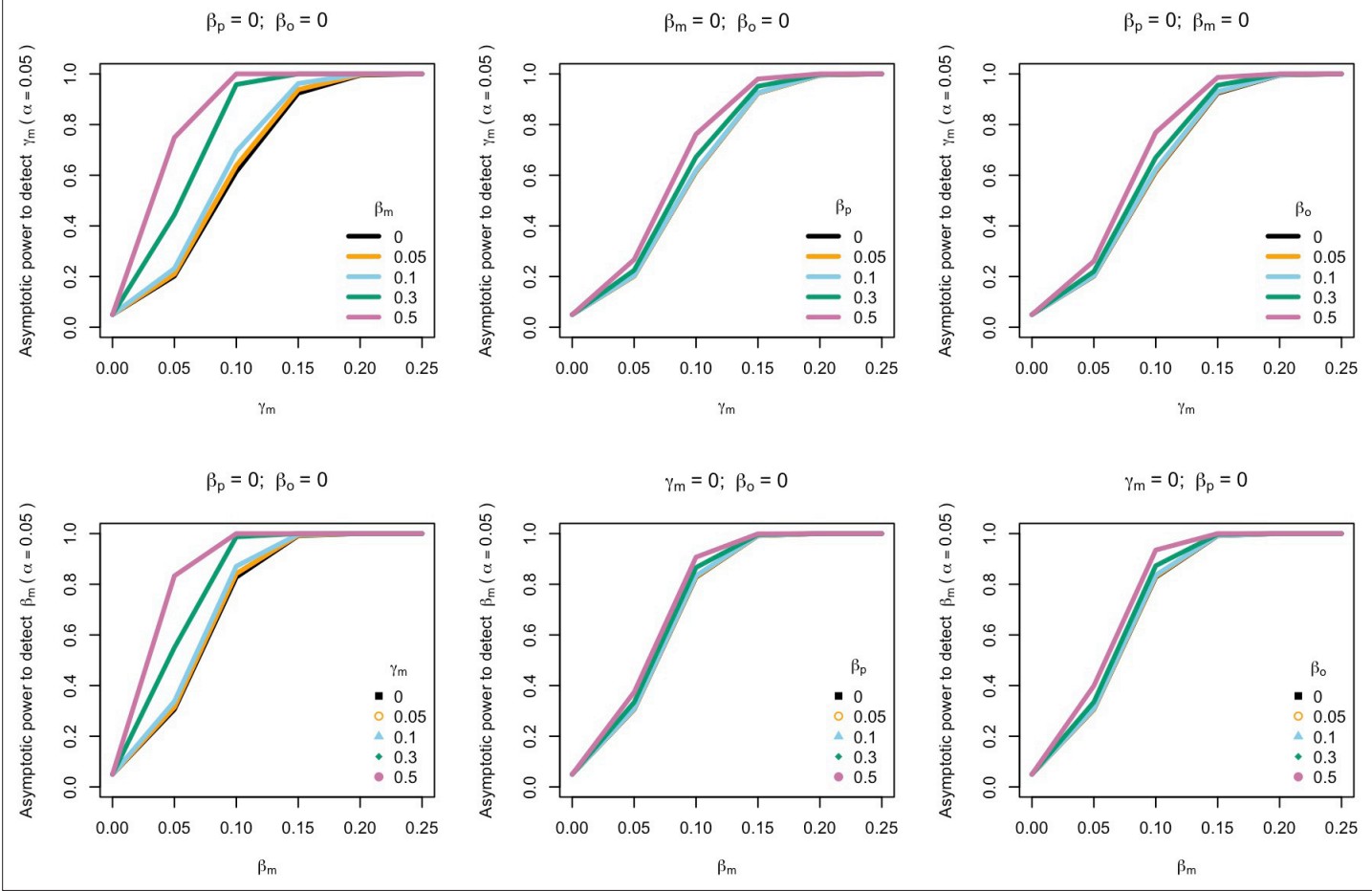

**Figure 2.** Power to detect prenatal maternal genetic effects ($\gamma_m$) (top) or postnatal maternal genetic effects ($\beta_m$) (bottom) whilst also varying the size of prenatal and postnatal maternal genetic effects, paternal genetic effects ($\beta_p$) or offspring genetic effects ($\beta_o$). Effect sizes are parameterized using the path coefficients $\beta$ and $\gamma$. Power was calculated assuming sample sizes approximating the number of white European individuals in the UK Biobank with educational attainment data (i.e. 1000 biological trios, 4000 biological mother-offspring pairs, 1800 biological father-offspring pairs, 300,000 singletons, 6000 adopted individuals, and 50 biological mother-adopted offspring pairs and a covariance of 0 ($\rho$) between maternal and paternal genotypes).

adopted singletons where 0%, 20%, 40%, 60%, or 80% had adoptive mother who were siblings or cousins of biological fathers, (iii) adopted singletons where 0%, 20%, 40%, 60%, or 80% had adoptive fathers who were siblings or cousins of biological fathers, and (vi) adopted singletons where 0%, 20%, 40%, 60%, or 80% had adoptive fathers who were siblings or cousins of biological mothers. We assumed that sample sizes and the number of individuals in the other groups were the same as in our asymptotic power calculations.

We further investigated type 1 error rates and estimates of prenatal and postnatal maternal genetic effects after correctly modelling the relationship between biological and adoptive parents. We created four additional family structures (G8 to G11, *Appendix 1—figure 1*) considering the above four scenarios by adding a covariance path (*r*) between the genotypes of adoptive and biological parents. The covariance between the genotypes of adoptive and biological parents was fixed to $r=0.5 \times \Phi$ when the adoptive parent was a sibling of the biological parent and $r=0.125 \times \Phi$ when they were cousins.

# Results
## Power calculations

The power to detect prenatal maternal genetic effects increased with increasing size of postnatal maternal genetic effects, paternal genetic effects, and offspring genetic effects (*Figure 2*). A similar

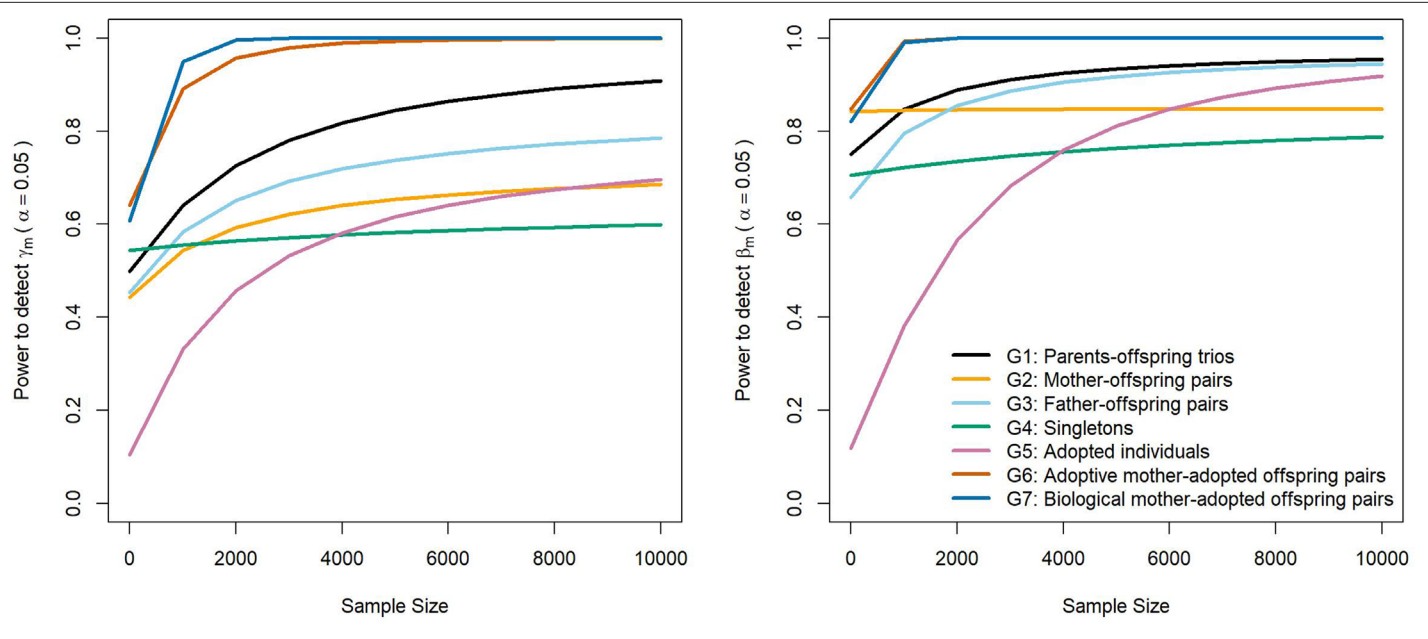

**Figure 3.** Power to detect prenatal maternal ($\gamma_m$) or postnatal maternal genetic ($\beta_m$) effects whilst varying the numbers of each family structure, with the sample sizes of other family structures approximating numbers of white European individuals in the UK Biobank reporting educational attainment (1000 biological trios, 4000 biological mother-offspring pairs, 1800 biological father-offspring pairs, 300,000 singletons, 6000 singletons, and 50 biological mother-adopted offspring pairs). Path coefficients representing postnatal or prenatal maternal genetic effects, paternal genetic effects ($\beta_p$) and offspring genetic effects ($\beta_o$) were fixed to 0.1. The covariance between maternal and paternal genotypes was fixed to 0.

pattern was observed when we quantified the power to detect postnatal maternal genetic effects. Our calculations indicate that a study with a sample size and structure similar to the UK Biobank would have ~80% power ($\alpha$=0.05) to detect prenatal maternal and postnatal maternal genetic effects that account for 1.5% and 0.9% of the variance in the offspring phenotype, respectively (calculated as $\gamma_m^2$ and $\mathscr{6}_m^2$ assuming all the other genetic effects are null and zero correlation between parental genotypes). The power to detect prenatal and postnatal maternal genetic effects also increased slightly with increasing covariance between maternal and paternal genotypes (*Appendix 1—figure 2*).

The power to detect prenatal and postnatal maternal genetic effects increased with the number of adopted singleton individuals, and more dramatically when the number of adoptive mother-adopted offspring pairs and/or biological mother-adopted offspring pairs increased (*Figure 3*). Power to detect prenatal maternal genetic effects increased most rapidly with increasing numbers of biological-mother adopted child pairs (G7), whereas the power to detect postnatal maternal genetic effects increased most strongly with increasing numbers of was provided by adoptive mother-adopted child pairs (G6). This makes sense intuitively, since adoptive mother-adopted child pairs (G6) provide a direct estimate of the postnatal maternal genetic effect ($\mathscr{6}_m$), whereas biological mother-adopted child pairs provide a direct estimate of the prenatal maternal genetic effect ($\gamma_m$). Likewise, adopted singleton individuals do not provide direct estimates of either prenatal or postnatal maternal genetic effects, and therefore contribute proportionally less in terms of statistical power than adoptive mother-adopted child or biological mother-adopted child pairs.

Interestingly, the power to detect prenatal and postnatal maternal genetic effects also increased with the number of biological families in the analysis (G1 to G4) (*Figure 3*); however, this increase asymptoted with maximal possible power depending primarily on the number of adopted individuals in the analysis. This result suggests that power to partition maternal genetic effects can be optimized through having large numbers of biological relatives in the analysis, but cannot be increased beyond a level that depends on the number of adopted individuals in the analysis (G5 to G7) and may be less than 100%. Results from simulations showed good concordance with asymptotic power calculations and appropriate type I error rates (*Supplementary file 3*, *Supplementary file 4*).

**Table 1.** Modelling results of birth weight in the UK Biobank.

| | Unweighed PRS | | | Weighted PRS | | |
|---|---|---|---|---|---|---|
| | Estimate | Std Error | P-value | Estimate | Std Error | P-value |
| Offspring effect | −0.008 | 0.004 | 0.058 | −0.173 | 0.115 | 0.132 |
| Prenatal maternal effect | 0.035 | 0.013 | 0.006 | 0.893 | 0.383 | 0.020 |
| Postnatal maternal effect | −0.017 | 0.011 | 0.130 | −0.419 | 0.335 | 0.211 |
| Postnatal paternal effect | 0.003 | 0.005 | 0.588 | 0.048 | 0.151 | 0.751 |

PRS: polygenic risk score constructed using 20 SNPs showing maternal effects on birth weight from **Warrington et al., 2019**; Std Error: standard error.

## Analyses in the UK Biobank

PRS for birth weight exhibited significant evidence of prenatal maternal genetic effects, offspring genetic effects (in the opposite direction), and no significant evidence for postnatal maternal or paternal genetic effects (**Table 1**). This pattern of results is expected since (a) birth weight is known to be affected by both maternal and fetal genetic effects, (b) maternal and offspring genetic effects on birth weight for many of the SNPs were in opposite directions, (c) conditional estimates of maternal and offspring genetic effects are negatively correlated, (d) in the present study SNPs comprising the PRS were selected for their strong maternal effects on offspring birth weight, and (e) birth weight is a perinatal phenotype and so by definition cannot be influenced by postnatal maternal (or paternal) genetic effects (**Warrington et al., 2019**; **Beaumont et al., 2018**; **Warrington et al., 2018**).

In the case of offspring educational attainment, our analyses showed evidence for a prenatal maternal genetic effect, a paternal genetic effect, and an offspring genetic effect (**Table 2**, **Supplementary file 5**). The absence of a significant positive postnatal maternal genetic effect on offspring educational attainment was surprising given that maternal genetic effects should be mediated through maternal educational attainment, and therefore should mostly involve postnatal pathways (although it is possible that some of the relationship between maternal genotype and offspring educational attainment could be mediated through prenatal effects- e.g. less educated mothers consuming more alcohol during pregnancy which then has adverse effects on offspring cognitive development and

**Table 2.** Modelling results of educational attainment in the UK Biobank.

| | Full sample | | | | | |
|---|---|---|---|---|---|---|
| | Unweighted PRS | | | Weighted PRS | | |
| | Estimate | Std Error | p-value | Estimate | Std Error | p-value |
| Offspring effect | 0.026 | 0.005 | $3.18 \times 10^{-7}$ | 2.474 | 0.485 | $3.40 \times 10^{-7}$ |
| Prenatal maternal effect | 0.027 | 0.011 | 0.013 | 1.722 | 0.703 | 0.014 |
| Postnatal maternal effect | −0.006 | 0.008 | 0.484 | −0.288 | 0.514 | 0.575 |
| Postnatal paternal effect | 0.018 | 0.007 | 0.006 | 1.233 | 0.678 | 0.069 |
| | **Excluding adopted individuals with breastfeeding information** | | | | | |
| | **Unweighted PRS** | | | **Weighted PRS** | | |
| | Estimate | Std Error | p-value | Estimate | Std Error | p-value |
| Offspring effect | 0.025 | 0.005 | $5.52 \times 10^{-7}$ | 2.441 | 0.372 | $5.15 \times 10^{-11}$ |
| Prenatal maternal effect | 0.017 | 0.012 | 0.140 | 0.909 | 0.705 | 0.197 |
| Postnatal maternal effect | 0.004 | 0.009 | 0.682 | 0.549 | 0.595 | 0.357 |
| Postnatal paternal effect | 0.019 | 0.007 | 0.005 | 1.273 | 0.552 | 0.021 |

PRS: polygenic risk score constructed using 1,267 SNPs from **Lee et al., 2018**; Std Error: standard error.

educational attainment etc), and previous studies have shown that the effect of paternal and maternal PRS on educational attainment are roughly similar (*Wang et al., 2021*) that is suggesting the absence of prenatal maternal genetic effects. We were concerned that the cause of this surprising result might be because of model misspecification- specifically, adopted children being raised by adoptive parents who are biologically related to them.

We therefore performed sensitivity analyses where we excluded adopted individuals who reported breastfeeding information (i.e. on the hypothesis that knowing if they were breast fed or not is information that an adopted individual could only know if they were adopted by a relative). The results of these analyses showed that estimates of the prenatal maternal genetic effect reduced in size and became non-significant. Whilst the postnatal maternal effect remained non-significant, the direction of effect changed from negative to positive. Sensitivity analyses using PRS constructed using the 72 genome-wide significant SNPs from the *Okbay et al., 2016* GWAS were underpowered but showed similar directions of effect *Okbay et al., 2016*.

## Simulations where adoptive parents are related to biological parents

Our simulations showed that the presence of adoptive parents who were genetically related to their adopted offspring could in some cases increase the type 1 error rate to detect prenatal maternal genetic effects and bias estimates of prenatal and postnatal maternal genetic effects when these relationships were not accurately modelled in the SEM (*Appendix 1—figure 3*, *Appendix 1—figure 4*, *Appendix 1—figure 5*, *Appendix 1—figure 6*). In general, the effect of including unmodelled related adoptive and biological parents in the SEM depended on whether the adoptive mother or adoptive father was related to the biological parents (i.e. it did not matter whether the adoptive parents were related to the biological mother or father) and the degree of relatedness (i.e. closer relationships had the potential to produce greater bias and type I error rates).

In the case of adoptive mothers being genetically related to biological parents (*Appendix 1—figure 3*, *Appendix 1—figure 4*), the presence of postnatal genetic effects (i.e. $\beta_m$ not equal to 0) was sufficient to bias estimates of the prenatal maternal genetic effect ($\gamma_m$) and increase type I error rates. In general, when $\beta_m$ was not equal to zero, then estimates of $\gamma_m$ were biased towards the true total maternal genetic effect (i.e. $\gamma_m + \beta_m$), and estimates of $\beta_m$ were biased towards zero. The reason for this can be seen intuitively by examining the path models for the G8 and G9 family structures in *Appendix 1—figure 1* where the presence of biologically related adoptive mothers leads to an unmodelled covariance path between offspring genotype and phenotype (i.e. $\frac{1}{2}r \times \beta_m$). Note that the expected covariance implied by this path is the same in the case of adoptive mothers who are related to their adopted offsprings' biological mothers (G8 in *Appendix 1—figure 1*), and adoptive mothers who are related to their adopted offsprings' biological fathers (G9 in *Appendix 1—figure 1*)- implying that failure to model relatedness in both groups should produce the same consequences in terms of bias and type I error. When $\beta_m$ is not zero, this unmodelled path alters the expected covariance between offspring genotype and phenotype in adopted singleton individuals. In this situation, the model will incorrectly attribute the altered covariance to prenatal maternal genetic effects (i.e. since the $\frac{1}{2}r \times \beta_m$ path is not explicitly modelled), and since estimates of the total maternal genetic effect remain unbiased (data not shown), will bias estimates of the postnatal maternal genetic effect towards zero. It follows that results will not be biased when $\beta_m = 0$ and the magnitude of any bias will decrease as $r$ approaches zero (i.e. with decreasing genetic relationship between adoptive mothers and the biological parents).

Likewise, our simulations showed that the presence of paternal genetic effects (i.e. $\beta_p$ not equal to 0) was sufficient to bias estimates of the prenatal maternal genetic effect ($\gamma_m$) and increase type I error rates when unmodelled adoptive fathers who were genetically related to adopted children's biological parents were included in the analysis (*Appendix 1—figure 5*, *Appendix 1—figure 6*). Inspection of the G10 and G11 family structures in *Appendix 1—figure 1* shows that biologically related adoptive fathers leads to an extra unmodelled covariance path between offspring genotype and phenotype (i.e. $\frac{1}{2}r \times \beta_p$), and that this path is the same for both adoptive fathers related to biological fathers (G10 in *Appendix 1—figure 1*) and adoptive fathers related to biological mothers (G11 in *Appendix 1—figure 1*) (again implying that failure to model relatedness in both groups should have the same consequences). It also shows that there should be no bias/inflation of type I error rates in the absence of paternal genetic effects, which is consistent with the results from our simulations. However, if $\beta_p$

is not zero, then this will affect the covariance between (adopted) offspring genotype and phenotype, and changes may be falsely ascribed to prenatal maternal genetic effects. In this situation, estimates of the prenatal maternal genetic effect ($\gamma_m$) will be biased towards the sum of the true prenatal maternal genetic effect plus the true paternal genetic effect ($\gamma_m + \beta_p$), and estimates of the postnatal maternal genetic effect ($\beta_m$) will be biased towards the difference between the true total maternal genetic effect ($\gamma_m + \beta_m$) minus the estimated prenatal maternal genetic effect ($\gamma_m$).

In general, the presence of unmodelled relationships between biological and adoptive parents did not bias estimates of offspring genetic or paternal genetic effects (results not shown). This makes sense since offspring and paternal genetic effects are estimated directly from the covariance between observed genotypes and phenotypes (i.e. the covariance between observed offspring genotype and offspring phenotype, and the covariance between observed paternal genotype and offspring phenotype, respectively), whereas the estimation of prenatal and postnatal maternal genetic effects have to be inferred indirectly using latent (not directly observed) genotypes and the difference in the covariance between offspring genotype and phenotype in biological and adoptive families.

When we correctly modelled the relationship between biological and adoptive parents' genotypes, there was no inflation in type 1 error rates and effect estimates were unbiased (*Appendix 1—figure 7*, *Appendix 1—figure 8*, *Appendix 1—figure 9*, *Appendix 1—figure 10*). However, correct modelling and inclusion of these individuals in the SEM produced complicated effects on the power to detect true effects relative to if the same number of adopted individuals with biologically unrelated adoptive parents had been included in the model (see *Appendix 1—figure 11* where 0% on the x-axis corresponds to this situation). For the simulations we examined, power to detect prenatal and postnatal maternal genetic effects decreased when adopted singletons whose adoptive mothers were related to their biological parents were included (and modelled correctly) in the analysis. This makes sense intuitively in that correct modelling of these mothers requires a covariance path between adoptive mother's genotype and the relevant biological parent's genotype (G8, G9). The presence of this path decreases the difference between the expected covariance between an adopted offspring's genotype and phenotype (G8, G9), and the expected covariance between offspring genotype and phenotype from a non-adoptive family (G4)- hence lowering the power of the model to discriminate between maternal prenatal and postnatal genetic effects.

In contrast, power to detect prenatal and postnatal maternal genetic effects increased when adopted singletons whose adoptive fathers were related to their biological parents were included (and modelled correctly) in the analysis. In order to understand this result intuitively, it is useful to consider the case where the offspring trait is affected by concordant prenatal and postnatal maternal genetic effects (i.e. $\gamma_m > 0$ and $\beta_m > 0$), but not by paternal ($\beta_p = 0$) or offspring genetic effects ($\beta_o = 0$). Under the full model, adopted singletons (G5) and adopted singletons whose adoptive father is a genetic relative (G10, G11) produce identical expected covariance matrices and so lead to identical model fits and parameter estimates. However, under the reduced model where $\gamma_m$ is fixed to zero, the only path that accounts for the covariance between offspring genotype and phenotype is the offspring genetic effect ($\beta_o$). Thus in order for this covariance to be modelled accurately, the reduced model will produce positive estimates for the offspring genetic effect ($\beta_o$). Simultaneously, to ensure that the covariance between the observed paternal genotype and offspring phenotype remains close to zero in parent-offspring trios and father-offspring pairs (i.e. G1 and G3), the model will produce negative estimates of the paternal genetic effect ($\beta_p$). This process presents more of a challenge when modelling adopted singleton individuals whose adoptive fathers are related to their biological parents (i.e. G10, G11) where the paternal genetic effect impacts both the expected covariance between parental genotype and offspring phenotype, and the expected covariance between offspring genotype and phenotype (i.e. because of the presence of the fixed covariance path *r*). The consequence is that the reduced model doesn't fit the data as well and increased power to detect association. Similar thought experiments can be used to provide intuition for the other results in *Appendix 1—figure 11*.

## Discussion

Genetic studies of adopted individuals have a venerable history in the field of behavior genetics (*Leahy, 1935*; *Richardson, 1913*). Traditionally, adoption studies were used to determine the relative importance of genes and the environment on trait variability- the basic idea being that similarity between adopted individuals and their biological parents reflects the effect of genes, whereas the

similarity between adopted individuals and their adoptive parents reflects the effect of shared environmental influences. Indeed, large studies of adopted individuals played a valuable role in resolving the 'Nature vs Nurture' debate of last century, complementing similar findings from twin studies and other pedigree-based designs (*Horn, 1983*; *Rhea et al., 2013*; *Scarr and Weinberg, 1983*).

In this manuscript, we developed a new statistical model that uses the information provided by adopted individuals to partition maternal genetic effects into prenatal and postnatal influences on offspring phenotypes. Our method uses SEM to model the expected covariance between genotypes and phenotypes in adopted and non-adopted individuals. We have specifically designed our method to capitalize on the large number of adopted 'singleton' individuals present in large-scale population-based resources like the UK Biobank, exploiting the fact that the expected covariance between offspring genotype and phenotype differs between adopted and non-adopted individuals as a function of postnatal maternal genetic (and other) effects. Whilst our method does not provide explicit mechanistic insight into the reason for the existence of maternal genetic effects (merely a simple partitioning into prenatal and postnatal components, and estimation as to the relative importance of each), follow up studies could help elucidate these mechanisms by for example examining the association between maternal PRS and different prenatal and postnatal maternal phenotypes, and performing mediation and Mendelian randomization analyses (*Evans et al., 2019*; *Armstrong-Carter et al., 2020*).

We are not the first to note that information from adopted individuals could be used to provide valuable information on the effect of the prenatal environment on offspring traits (*Loehlin, 2016*; *Demange et al., 2020*). We are the first, however, to our knowledge to develop a model that specifically quantifies prenatal and postnatal maternal genetic effects on offspring phenotypes. Recently, (*Domingue and Fletcher, 2020*) constructed a PRS to proxy educational attainment using maternal genotypes, and tested the strength of the association between the risk score and offspring educational attainment in children from biological and adoptive families separately (*Domingue and Fletcher, 2020*). The authors showed that the positive association between the score and educational attainment was stronger in children from biological than adoptive families. Our method differs from Dominique and Fletcher in a number of important respects. First, the focus in their study was on demonstrating indirect parental effects on offspring phenotypes, whereas the focus in our manuscript is specifically on partitioning maternal genetic effects into prenatal and postnatal effects. Second, our method provides point estimates and tests of the significance of prenatal and postnatal maternal genetic effects. Lastly, the method used in Dominique and Fletcher requires both adopted and biological parent-child trios (or alternatively mother-offspring pairs). In contrast, our method at a minimum requires biological parent-child trios (or both biological mother-offspring pairs and biological father-offspring pairs) and adopted singletons only to estimate prenatal and postnatal maternal genetic effects on offspring phenotypes. The inclusion of adoptive mother-adopted offspring pairs and/or biological mother-adopted offspring pairs increases power to partition maternal genetic effects dramatically, but is not a strict requirement of our method if adopted singletons are available.

We are also not the first to have used adopted individuals in the UK Biobank to estimate the contribution of indirect genetic effects to phenotypic variability. (*Cheesman et al., 2020*) contrasted adopted and non-adopted individuals in the UK Biobank to investigate evidence for indirect genetic effects on educational attainment and the existence of gene-environment correlation (*Cheesman et al., 2020*). The authors found that a genome-wide PRS for educational attainment was more strongly associated with educational attainment in non-adopted compared to adopted individuals consistent with the existence of indirect genetic effects on educational attainment. Likewise, G-REML estimates of SNP heritability were also lower in adopted individuals compared to non-adopted individuals (potentially due to the absence/lower amount of passive gene-environment correlation in adopted individuals).

In a previous study of cognitive and non-cognitive effects on educational attainment (*Demange et al., 2020*), used three different sorts of study design to estimate the contribution of indirect genetic effects on educational attainment (*Demange et al., 2020*). They estimated indirect genetic effects by comparing between family and within family genetic effects in sibling pairs (*Fulker et al., 1999*), by estimating the effect of non-transmitted parental alleles on offspring phenotype in parent-offspring trios, and by computing the difference in PRS associations between adopted and non-adopted individuals. In general, the authors found that estimates of indirect effects using the adoption study design were lower than those produced by the other two models. They interpreted this as being

a consequence of the adoption study methodology not capturing the impact of prenatal maternal genetic effects (and also potentially being more robust to assortative mating and population stratification) in contrast to the two other study designs. The present study explicitly capitalizes on differences between adopted and non-adopted individuals in their pattern of genotype-trait covariances to estimate the relative size of prenatal and postnatal maternal genetic effects and to provide formal statistical tests for their presence. Indeed, both (*Demange et al., 2020*) and our findings are consistent with the existence of prenatal effects on educational attainment (although we believe that at least part of this finding is an artefact and due to unmodelled biological relationships between some adopted children and their adoptive parents in the UK Biobank).

Our power calculations show that in principle the number of adopted individuals in the UK Biobank is unlikely to be sufficient to resolve effects at individual genetic loci but may be large enough to detect evidence for prenatal/postnatal maternal genetic effects in the case of PRS. It is worth noting that, whilst primarily a function of the number of adoptive families in the analysis, the power to partition maternal genetic effects was also influenced by the number of biological families analysed. Intuitively, this is because the other family structures contribute to more precise estimation of for example paternal genetic effects, which in turn provides information on the total size of the maternal genetic component. This observation is important, because the number of adopted individuals in an analysis is likely to be severely constrained, whereas power can sometimes be improved (at least up to a threshold determined by the total number of adoptive families/individuals in the analysis) by increasing the sample size of other relative types that may be easier to ascertain. Regardless, the most efficient way to increase the power to detect prenatal and/or postnatal maternal genetic effects is to genotype adoptive mother-adopted child pairs and/or biological mother-adopted child pairs.

Our empirical analyses in the UK Biobank showed the expected pattern of maternal genetic effects on offspring birth weight (i.e. strong evidence for prenatal but not postnatal maternal genetic effects) but not on offspring educational attainment. Parental educational attainment is known to causally affect offspring educational attainment (*Bates et al., 2018*; *Kong et al., 2018*; *Hwang et al., 2020*). Whilst it is possible, even probable, that some loci may affect offspring educational attainment through prenatal maternal pathways, it is not credible that such mechanisms would lead to larger effect sizes than those from postnatal pathways, or that fathers would exhibit strong evidence of postnatal effects but not mothers. Rather we hypothesize it is possible that our model could have been misspecified in that substantial numbers of adopted individuals in the UK Biobank may have in fact been raised by their biological relatives. This can be thought of as (unintentional) reintroduction of passive gene-environment correlation into the study. In other words, adopted children are brought up by their genetic relatives, who in turn provide the environment in which they are raised. This induces a correlation between adopted individuals' PRS and their environment. Sensitivity analyses that excluded adopted individuals with known breastfeeding information suggested that including adopted individuals who were raised by genetically related adoptive mothers may lead to inflated prenatal maternal genetic effect estimates. Our simulations with adoptive parents who were siblings or cousins of the biological parents further confirmed that unmodelled relatedness can increase evidence for prenatal maternal genetic effects at the expense of postnatal effects, particularly when paternal genetic and postnatal maternal effects are present, as would be the case with educational attainment (this contrasts with birth weight where we do not expect any paternal or postnatal genetic effects and for which we obtained sensible results). Ideally, one would have detailed information on any genetic relationship between adoptive and biological parents and model these relationships in the SEM (see below).

Our model relies on several strong assumptions, some of which are shown explicitly in *Figure 1* and others which are not (see *Supplementary file 6* for a summary of some of the major assumptions/limitations of our model). Those assumptions explicitly encoded in *Figure 1* include that the total maternal genetic effect can be decomposed into the sum of prenatal and postnatal components, that genetic effects are homogenous across biological and adoptive families, the absence of genotype x environment interaction, the absence of gene-environment correlation (including evocative gene-environment correlation and correlation between the prenatal maternal environment and maternal PRS), any correlation between the genetic scores in spouses is the same in biological and adoptive families, and that fathers do not exert prenatal effects on offspring traits. Assumptions that are not explicitly shown in *Figure 1* include that adopted (and adoptive) individuals are not systematically

different from other individuals, adopted individuals are placed randomly within the population (including with individuals not genetically related to themselves), that adoption happened soon after birth and that the adopted individuals were raised in one adoptive family, and adopted individuals do not maintain contact with their biological parents. We also ignore any complexities due to for example single mothers raising offspring independently of their fathers etc.

Whilst we expect some of these assumptions to hold (i.e. it is reasonable to expect the absence of substantial prenatal paternal genetic effects for many traits), we expect that others will show varying degrees of violation with differing consequences for the parameter estimates obtained under the misspecified model. For example, it is well appreciated that adopted individuals and adoptive families may differ in several respects to the general population. For example, some studies have reported that adoptive families are better educated and have higher socio-economic status than the population average (*Kendler et al., 2015*). We argue that violation of this assumption is likely to exert more serious consequences on traditional adoption studies which model the phenotypic correlation between biological, adopted relatives. In contrast, in our design it is more important that genetic effect sizes are homogenous across adopted and non-adopted individuals (i.e. no genotype by environment interaction), and at least currently, there is limited evidence for large genotype by environment interactions at individual genetic loci. Our model also allows for differences in the residual variance across biological and adoptive families (*Cheesman et al., 2020*). This may be important if for example, adopted children are selectively placed in a reduced range of environments compared to their non-adopted counterparts.

We argue that of greater consequence for the validity of our model is that any genetic relationship between adoptive and biological parents is accurately modelled and included in the SEM. Through simulation, we have shown that the consequences of model misspecification depend upon which biological and adoptive parents are related, the nature of this relationship, and the proportion of adopted individuals in the sample who have had their relationship misspecified. Our simulations also showed that correctly modelling this relationship returns asymptotically unbiased effect estimates and correct type I error rates. Clearly, knowing these cryptic relationships in the UK Biobank would allow us to properly model them and better estimate prenatal and postnatal maternal genetic effects using this resource. We emphasize that accurately modelling these relationships does not require that actual genotypes for adoptive and/or biological parents be obtained (although this would be advantageous in terms of statistical power) as our SEM allows us to model these relationships in terms of latent variables. Indeed, as large-scale resources like the UK Biobank become more common, we expect that the number of adopted individuals who have GWAS will also increase, and consequently models like the one espoused in this manuscript will become increasingly useful. High quality phenotypic information on these adopted individuals and their adoptive parents including whether they share any biological relationship will be critical to making the most of these resources.

We have also not modelled the complex effects of assortative mating in our SEM other than including covariance terms between maternal and paternal genotypes and assuming equal genetic variances in parents and their offspring under random mating (i.e. this is equivalent to assuming one round of assortative mating in the parental generation). Positive assortment induces a number of complications when attempting to decompose the offspring phenotypic variance into its constituent sources of variation, including increasing homozygosity and the genetic variance at loci for the trait undergoing assortment relative to that expected under Hardy-Weinberg equilibrium, and inducing correlations between assorting loci across the rest of the genome both within and between individuals in the same family. Spousal correlations for phenotypes (educational attainment: $r=0.285$; birth weight: $r=0.076$) and PRS (educational attainment: $r=0.117$; birth weight: $r=0.068$) in complete parent-offspring trios in the UK Biobank suggest that assortative mating is likely present for educational attainment, and potentially to a lesser degree for (phenotypes correlated with) birth weight. Recent work by *Balbona et al., 2021* and *Kim et al., 2021* have shown how PRS of transmitted and non-transmitted alleles in parent-offspring trios can be used to estimate direct and indirect genetic effects on offspring phenotype and the variation attributable to the environmental influence of parents on offspring under phenotypic assortment (*Balbona et al., 2021*; *Kim et al., 2021*). It is possible that our basic model could be extended in a similar fashion to incorporate the effect of assortment and estimate some of these effects also.

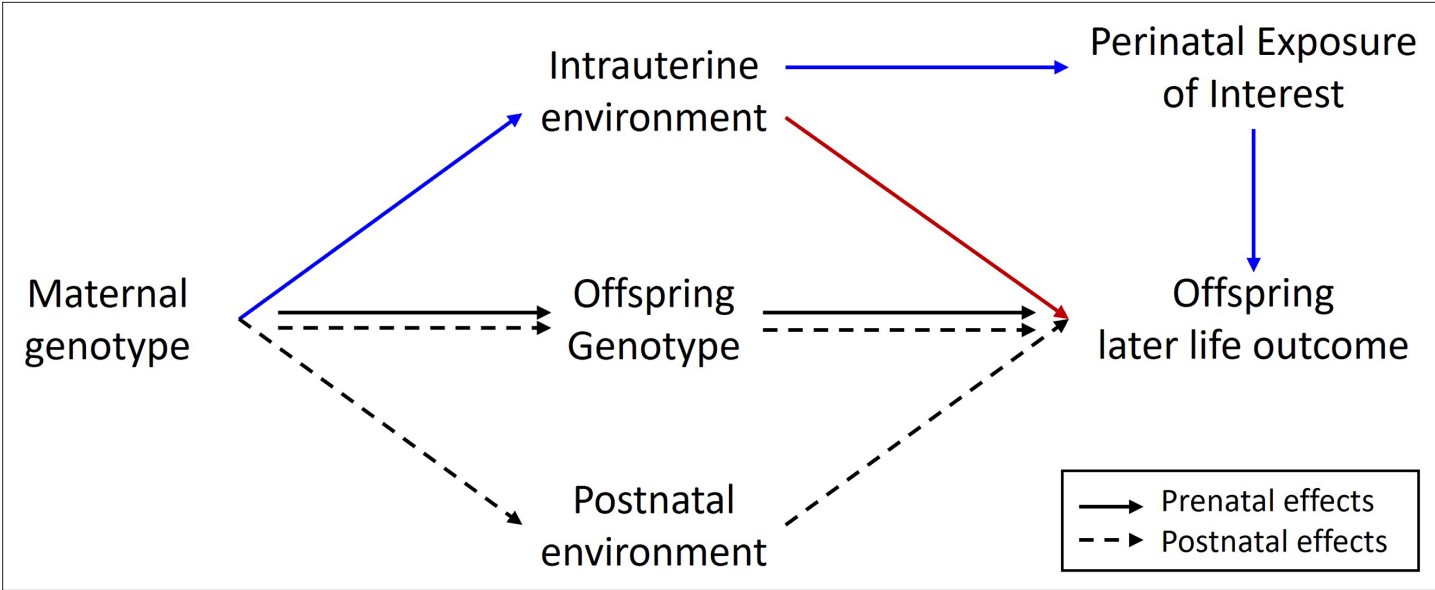

**Figure 4.** Diagram illustrating a Mendelian randomization study designed to estimate the causal effect of a maternal prenatal environmental exposure on an offspring later life outcome. Maternal genotype proxies a perinatal exposure of interest and is used as an instrumental variable to estimate the causal effect of the perinatal exposure on the offspring outcome. The pathway of interest is represented by the blue arrows where the SNP influences the outcome prenatally through the perinatal exposure of interest. In this sort of design, it is important to control for spurious pathways through the offspring genome since offspring genotype will be correlated ½ with maternal genotype. Maternal genotype could also influence the offspring phenotype via other pleiotropic paths through the intrauterine environment (red arrow) or through the postnatal environment (dashed arrows). The inclusion of adopted individuals into the research design may be useful in controlling for the effect of horizontal pleiotropic influences through the postnatal environment. Intuitively this is because adoptive mother's genotype provides an estimate of the relationship between maternal genotype and offspring outcome through postnatal pathways only. This estimate could be included in statistical models of the relationship between maternal genotype and offspring outcome to help correct for the effect of horizontal pleiotropy.

Also relevant to this discussion is a series of data simulations performed by *Demange et al., 2020* who found that estimates of indirect genetic effects obtained in adoption designs (i.e. which estimate the size of indirect genetic effects by taking the difference between a PRS-trait association in adopted individuals and a PRS-trait association in non-adopted individuals) were less biased by assortative mating and population stratification than estimates of indirect genetic effects obtained in biologically related parent-offspring trio designs (i.e. where estimates of indirect genetic effects are obtained by regressing offspring phenotype on parental genotype conditional on offspring genotype). In the case of population stratification, the intuition behind this observation is that the contaminating effects of substructure are present in both adopted and non-adopted individuals. This contamination cancels out when the PRS-trait association in adopted individuals is subtracted from the PRS-trait association in non-adopted individuals. However, this is not the case in parent-offspring trios, where estimates of indirect genetic effects remain contaminated.

In the context of our SEM, we expect that unmodelled population stratification will bias estimates of maternal prenatal and postnatal effects away from the null since population stratification will increase the covariance between maternal genotype and offspring phenotype, as well as the covariance between offspring genotype and phenotype in adopted and non-adopted individuals. Because we have limited our empirical analyses to individuals of white European ancestry and residualized the phenotypes for the first five genetic principal components prior to analysis in the SEM, any influence due to population stratification is likely to have been minor in our empirical analyses in the UK Biobank. The influence of assortative mating on our model is more difficult to predict, but our limited exploration of this issue by simulation (data not shown) suggests that the effect of assortative mating on parameter estimates from our SEM is likely to be minor.

Finally, it has not been lost on us, that our framework may provide a basis for correcting Mendelian randomization (MR) studies of intrauterine/early life environmental exposures for some of the contaminating effects of horizontal genetic pleiotropy (*Figure 4*). The presence of latent horizontal pleiotropy is one of the major threats to the validity of causal inference from MR studies (*Hemani*

*et al., 2018*). As we have shown, the inclusion of adopted individuals alongside biological families in genetic studies permits the partitioning of maternal genetic effects into prenatal and postnatal components. Consequently, if the focus of an MR study is on estimating the causal effect of a prenatal maternal exposure on an offspring outcome, the covariance between adoptive mother's genotype and offspring (outcome) phenotype should provide an estimate of genetic effects due to postnatal horizontal genetic pleiotropy. These estimates could then be included in an MR model to correct causal estimates for the effect of postnatal pleiotropy (*Figure 4*). It is important to realize, however, that this procedure will not correct for the effect of prenatal/early life horizontal pleiotropy as these paths will be present in both biological and adopted individuals and so this approach is not a panacea for dealing with latent pleiotropy in MR studies.

In conclusion, we present a simple extension to the basic adoption design that includes measured genotypes and enables partitioning of maternal genetic effects into prenatal and postnatal sources of variation. We show in principle that adopted singleton individuals in the UK Biobank combined with biologically related parent-offspring trios and pairs is sufficient to partition maternal genetic effects into prenatal and postnatal sources of variation. However, power calculations suggest that such a partitioning would currently only be realistic for PRS explaining substantial proportions of the variance in offspring phenotype, and that much larger numbers of individuals would be required to achieve such a partitioning at individual loci. In addition, failure to correctly model cryptic relationships between adoptive and biological parents may produce biased estimates of maternal genetic effects and increased type I error rates. Accurate modelling of cryptic relationships is sufficient to bring type I error rate under control and produce unbiased effect estimates. We suggest that there is a possibility that many individuals in the UK Biobank who report being adopted could have been raised by their biological relatives. We conclude that there would be considerable value in following up adopted individuals in the UK Biobank to determine whether they were raised by biological relatives, and if so, to precisely ascertain the nature of the relationship. These adopted individuals could then be used in informative statistical genetics models like the one described in the present manuscript to further elucidate the genetic architecture of complex traits and diseases.

## Code availability

R code for performing the analyses described in this manuscript is available in the Source Code File.

## Acknowledgements

This research has been conducted using the UK Biobank resource (Reference 53641). DME is funded by an Australian National Health and Medical Research Council Senior Research Fellowship (APP1137714) and this work was funded by NHMRC project grants (GNT1157714, GNT1183074). GHM is supported by the Norwegian Research Council (Post doctorial mobility research grant 287198) and Nils Normans minnegave. We would like to thank Mike Neale and Joshua Pritikin for discussions regarding the SEM.

## Additional information

### Funding

| Funder | Grant reference number | Author |
| --- | --- | --- |
| National Health and Medical Research Council | APP1137714 | David M Evans |
| National Health and Medical Research Council | GNT1157714 | David M Evans |
| National Health and Medical Research Council | GNT1183074 | David M Evans |
| Norwegian Research Council | 287198 | Gunn-Helen Moen |
| Nils Normans minnegave | | Gunn-Helen Moen |

| Funder | Grant reference number | Author |
|---|---|---|

The funders had no role in study design, data collection and interpretation, or the decision to submit the work for publication.

## Author contributions

Liang-Dar Hwang, Data curation, Software, Formal analysis, Validation, Investigation, Visualization, Methodology, Writing – original draft, Writing – review and editing; Gunn-Helen Moen, Data curation, Software, Validation, Writing – review and editing; David M Evans, Conceptualization, Resources, Data curation, Software, Formal analysis, Supervision, Funding acquisition, Investigation, Visualization, Methodology, Writing – original draft, Project administration, Writing – review and editing

## Author ORCIDs

Liang-Dar Hwang ⓘ http://orcid.org/0000-0002-5535-2199
David M Evans ⓘ http://orcid.org/0000-0003-0663-4621

## Ethics

Human subjects: The UK Biobank study was approved by the UK National Health Service National Research Ethics Service. Written consent was obtained from both the participants and their parents (for subjects younger than 18 years old). This study was approved by the Human Research Ethics Committee at the University of Queensland (approval number: 2019002705).

## Decision letter and Author response

Decision letter https://doi.org/10.7554/eLife.73671.sa1
Author response https://doi.org/10.7554/eLife.73671.sa2

## Additional files

### Supplementary files

• Supplementary file 1. SNPs used to construct unweighted polygenic scores of educational attainment.

• Supplementary file 2. SNPs used to construct unweighted polygenic scores of educational attainment.

• Supplementary file 3. Comparison of power estimated from asymptotic calculations and simulations using varying sizes of prenatal maternal genetic effects ($\gamma_m$), paternal genetic effects ($\beta_p$) or offspring genetic effects ($\beta_o$) with sample sizes approximating the number of individuals in UK Biobank with educational attainment information. 1000 biological parent-offspring trios, 4000 biological mother-offspring pairs, 1800 biological father-offspring pairs, 300,000 singletons, 6000 adopted individuals, and 50 biological mother-adopted offspring pairs. Covariance between parental genotypes was fixed at 0.

• Supplementary file 4. Comparison of power to detect pre-natal ($\gamma_m$) and post-natal ($\beta_m$) maternal genetic effects estimated from asymptotic calculations and simulations using varying sample size for each of the 7 family structures and the remaining family structures approximating the number of individuals in the UK Biobank with educational attainment data. 1000 biological parent-offspring trios, 4000 biological mother-offspring pairs, 1800 biological father-offspring pairs, 300,000 singletons, 6000 adopted individuals, and 50 biological mother-adopted offspring pairs. Path coefficients representing postnatal or prenatal maternal genetic effects, paternal genetic effects ($\beta_p$) and offspring genetic effects ($\beta_o$) were fixed to 0.1. Covariance between parental genotypes was fixed to 0.

• Supplementary file 5. Sensitivity analysis for the modelling of educational attainment in the UK Biobank using SNPs from the *Okbay et al., 2016*.

• Supplementary file 6. Some limitations/assumptions of our Structural Equation Model (SEM) and its application to the UK Biobank. Source Code File. R scripts that fit Structural Equation Modelling to partition maternal genetic effects into pre- and postnatal effects on offspring phenotypes.

• Transparent reporting form

• Source code 1. Adoption_SEM.R.

## Data availability

Human genotype and phenotype data on which the results of this study were based were accessed from the UK Biobank (http://www.ukbiobank.ac.uk/) with accession ID 53641. The genotype and phenotype data are available upon application from the UK Biobank (http://www.ukbiobank.ac.uk/). R code for performing the analyses described in this manuscript is available in the Supplementary Materials.

The following previously published dataset was used:

| Author(s) | Year | Dataset title | Dataset URL | Database and Identifier |
|---|---|---|---|---|
| Sudlow C, Gallacher J, Allen N, Beral V, Burton P, Danesh J, Downey P, Elliott P, Green J, Landray M, Lui B, Matthews P, Ong G, Pell J, Silman A, Young A, Sprosen T, Peakman T, Collins R | 2015 | UK Biobank | http://www.ukbiobank.ac.uk/ | UK Biobank, 53641 |

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

## Appendix 1

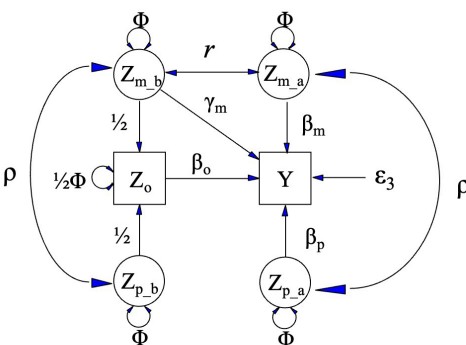

**G8: Adopted Individuals –
Biological and adoptive
mothers are related**

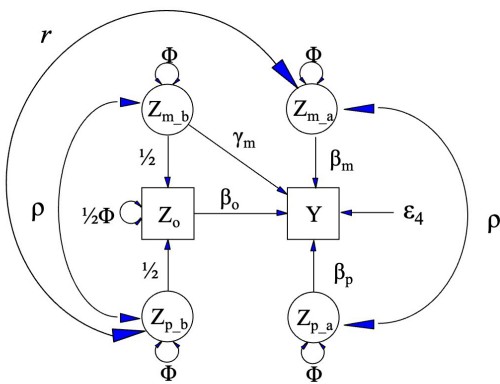

**G9: Adopted Individuals –
Biological father and
adoptive mother are related**

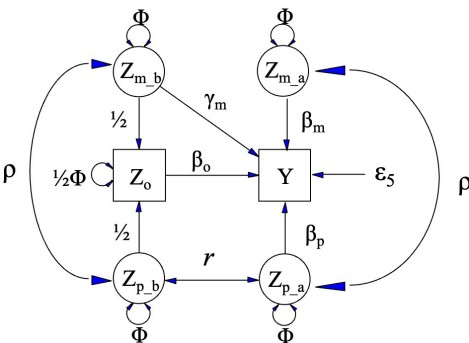

**G10: Adopted Individuals -
Biological and adoptive
fathers are related**

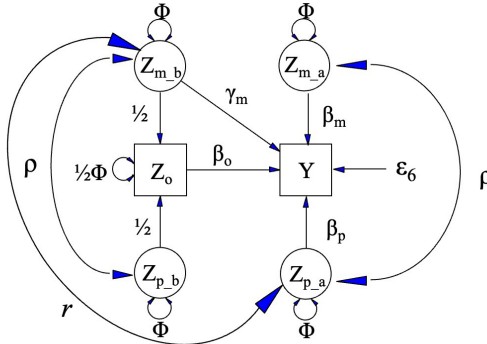

**G11: Adopted Individuals –
Biological mother and
adoptive father are related**

**Appendix 1—figure 1.** Path diagrams illustrating the structural equation models (SEM) underlying four additional family structures (G8 – G11) where biological and adoptive parents are related. Observed variables and latent variables are shown in squares and circles respectively. Causal relationships are represented by one headed arrows. Two headed arrows represents correlational relationships. $Z_o$ represents offspring genotype which influences offspring phenotype ($\beta_o$) and is correlated ½ with the genotypes of its biological parents. $Z_{m\_b}$ represents the genotype of a biological mother whose child was adopted and therefore only influences her child's phenotype through prenatal pathways ($\gamma_m$). $Z_{m\_a}$ represents the adoptive mother's genotype which only influences her adopted offspring's phenotype via postnatal pathways ($\beta_m$). $Z_{p\_b}$ represents the genotype of a biological father whose child was adopted and therefore has no influence on the adopted offspring phenotype. $Z_{p\_a}$ represents the adoptive father's genotype which influences his adopted offspring postnatally ($\beta_p$). $\rho$ represents the covariance between parental genotypes, as a result of e.g. assortative mating (it is assumed that this covariance is the same in biological parents and adoptive parents). The total variance of genotypes in the parental generation is set to $\Phi$. $\varepsilon_3$ through $\varepsilon_6$ represent residual error terms for the adopted offspring that we assume have different variances. The (fixed) parameter $r$ represents the covariance between biological and adoptive parents' genotypes.

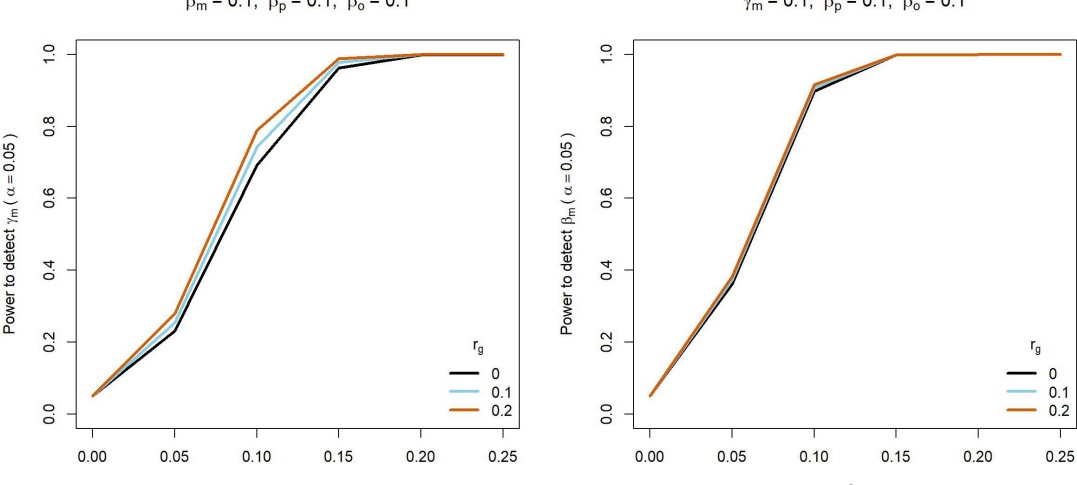

**Appendix 1—figure 2.** Power to detect prenatal maternal ($\gamma_m$) or postnatal maternal ($\beta_m$) genetic effects whilst varying the correlation ($r_g$) between maternal and paternal genotypes. Power was calculated using sample sizes approximating the number of white European individuals in the UK Biobank reporting their own educational attainment (1000 biological trios, 4000 biological mother-offspring pairs, 1,800 biological father-offspring pairs, 300,000 singletons, 6000 adopted individuals, and 50 biological mother-adopted offspring pairs). Path coefficients representing prenatal maternal ($\gamma_m$), postnatal maternal ($\beta_m$), paternal ($\beta_p$) and offspring genetic effects ($\beta_o$) were fixed to 0.1.

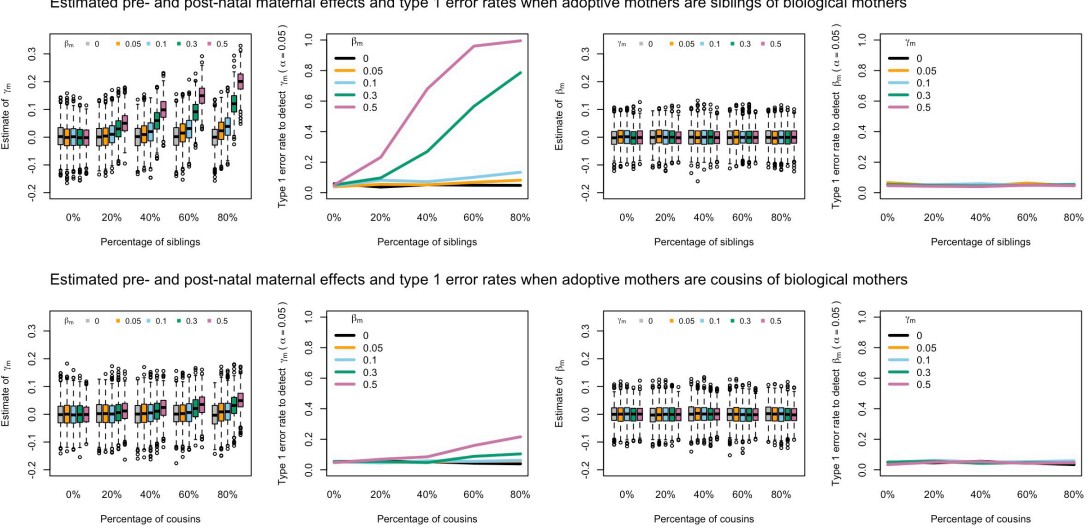

**Appendix 1—figure 3.** Estimated prenatal ($\gamma_m$) and postnatal ($\beta_m$) maternal effects and type 1 error rates whilst varying the percentage of adoptive mothers being siblings (top) or cousins (bottom) of biological mothers. An SEM which did not correctly model this relationship was fit to the data. Power was calculated using simulated data with 1000 biological trios, 4000 biological mother-offspring pairs, 1800 biological father-offspring pairs, 300,000 singletons, and 6000 adopted individuals. Prenatal maternal genetic effects were fixed to zero whilst the size of postnatal maternal genetic effects were varied (left panels); postnatal maternal genetic effects were fixed to zero whilst the size of prenatal maternal genetic effects were varied (right panels). Paternal effects ($\beta_p$) and offspring effects ($\beta_o$) were fixed to 0.1. The covariance between maternal and paternal genotypes was fixed to zero.

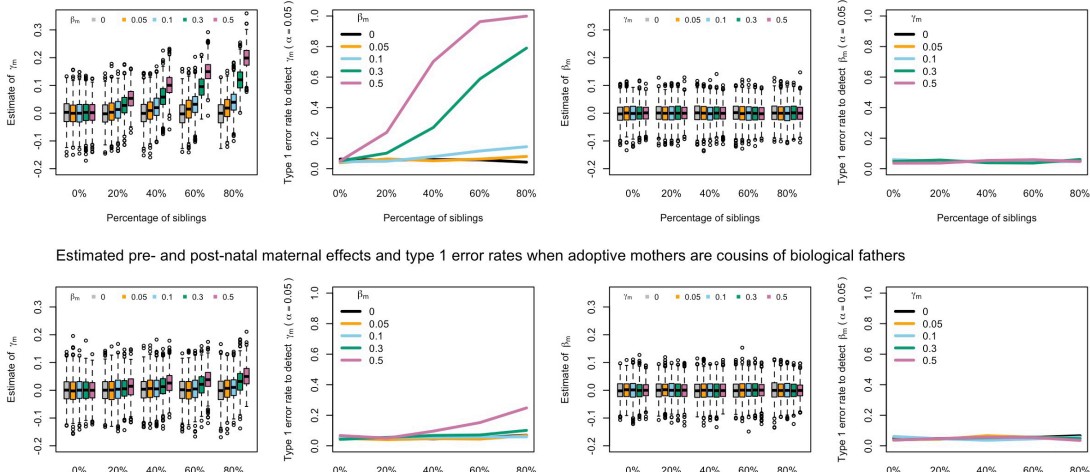

**Appendix 1—figure 4.** Estimated prenatal ($\gamma_m$) and postnatal ($\beta_m$) maternal effects and type 1 error rates whilst varying the percentage of adoptive mothers being siblings (top) or cousins (bottom) of biological fathers. An SEM which did not correctly model this relationship was fit to the data. Power was calculated using simulated data with 1000 biological trios, 4000 biological mother-offspring pairs, 1800 biological father-offspring pairs, 300,000 singletons, and 6000 adopted individuals. Prenatal maternal genetic effects were fixed to zero whilst the size of postnatal maternal genetic effects were varied (left panels); postnatal maternal genetic effects were fixed to zero whilst the size of prenatal maternal genetic effects were varied (right panels). Paternal effects ($\beta_p$) and offspring effects ($\beta_o$) were fixed to 0.1. The covariance between maternal and paternal genotypes was fixed to zero.

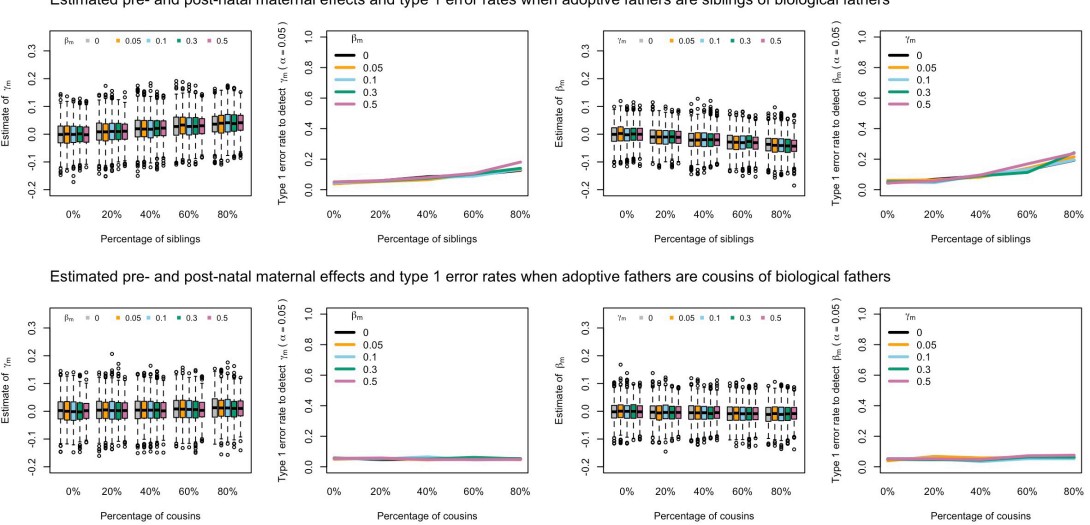

**Appendix 1—figure 5.** Estimated prenatal ($\gamma_m$) and postnatal ($\beta_m$) maternal effects and type 1 error rates whilst varying the percentage of adoptive fathers being siblings (top) or cousins (bottom) of biological fathers. An SEM which did not correctly model this relationship was fit to the data. Power was calculated using simulated data with 1000 biological trios, 4000 biological mother-offspring pairs, 1800 biological father-offspring pairs, 300,000 singletons, and 6000 adopted individuals. Prenatal maternal genetic effects were fixed to zero whilst the size of postnatal maternal genetic effects were varied (left panels); postnatal maternal genetic effects were fixed to zero whilst the size of prenatal maternal genetic effects were varied (right panels). Paternal effects ($\beta_p$) and offspring effects ($\beta_o$) were fixed to 0.1. The covariance between maternal and paternal genotypes was fixed to zero.

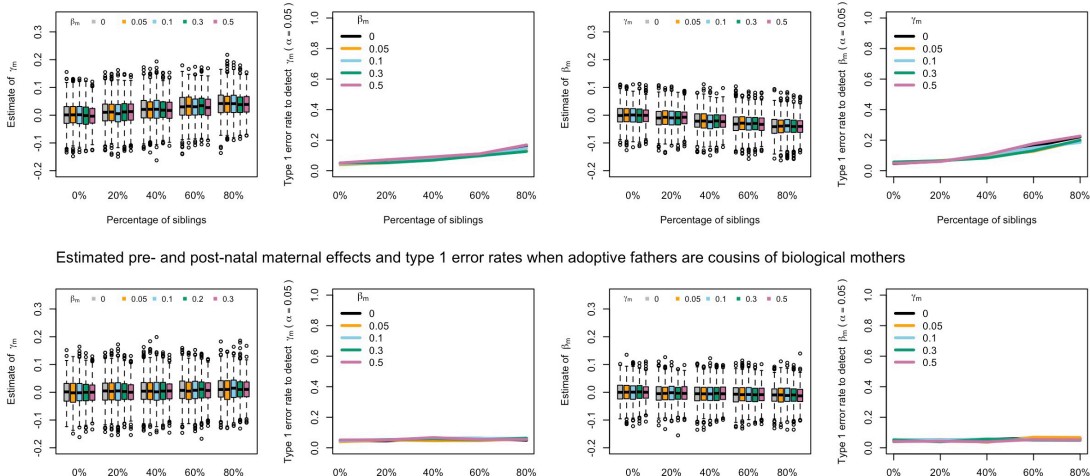

**Appendix 1—figure 6.** Estimated prenatal ($\gamma_m$) and postnatal ($\beta_m$) maternal effects and type 1 error rates whilst varying the percentage of adoptive fathers being siblings (top) or cousins (bottom) of biological mothers. An SEM which did not correctly model this relationship was fit to the data. Power was calculated using simulated data with 1000 biological trios, 4000 biological mother-offspring pairs, 1800 biological father-offspring pairs, 300,000 singletons, and 6000 adopted individuals. Prenatal maternal genetic effects were fixed to zero whilst the size of postnatal maternal genetic effects were varied (left panels); postnatal maternal genetic effects were fixed to zero whilst the size of prenatal maternal genetic effects were varied (right panels). Paternal effects ($\beta_p$) and offspring effects ($\beta_o$) were fixed to 0.1. The covariance between maternal and paternal genotypes was fixed to zero.

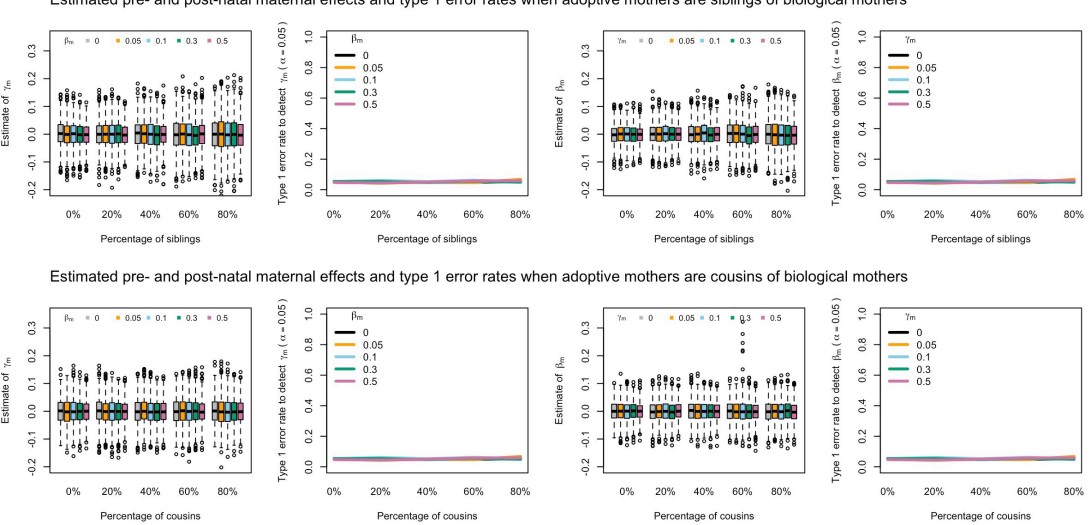

**Appendix 1—figure 7.** Estimated prenatal ($\gamma_m$) and postnatal ($\beta_m$) maternal effects and type 1 error rates whilst varying the percentage of adoptive mothers being siblings (top) or cousins (bottom) of biological mothers. An SEM which modelled this relationship correctly was fit to the data. Power was calculated using simulated data with 1000 biological trios, 4000 biological mother-offspring pairs, 1800 biological father-offspring pairs, 300,000 singletons, and 6000 adopted individuals. Prenatal maternal genetic effects were fixed to zero whilst the size of postnatal maternal genetic effects were varied (left panels); postnatal maternal genetic effects were fixed to zero whilst the size of prenatal maternal genetic effects were varied (right panels). Paternal effects ($\beta_p$) and offspring effects ($\beta_o$) were fixed to 0.1. The covariance between maternal and paternal genotypes was fixed to zero.

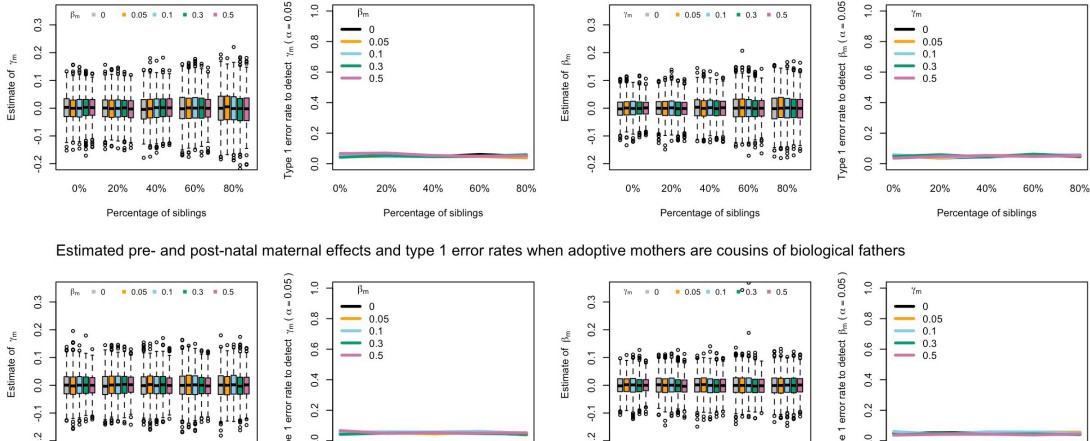

**Appendix 1—figure 8.** Estimated prenatal ($\gamma_m$) and postnatal ($\beta_m$) maternal effects and type 1 error rates whilst varying the percentage of adoptive mothers being siblings (top) or cousins (bottom) of biological fathers. An SEM which modelled this relationship correctly was fit to the data. Power was calculated using simulated data with 1000 biological trios, 4000 biological mother-offspring pairs, 1800 biological father-offspring pairs, 300,000 singletons, and 6000 adopted individuals. Prenatal maternal genetic effects were fixed to zero whilst the size of postnatal maternal genetic effects were varied (left panels); postnatal maternal genetic effects were fixed to zero whilst the size of prenatal maternal genetic effects were varied (right panels). Paternal effects ($\beta_p$) and offspring effects ($\beta_o$) were fixed to 0.1. The covariance between maternal and paternal genotypes was fixed to zero.

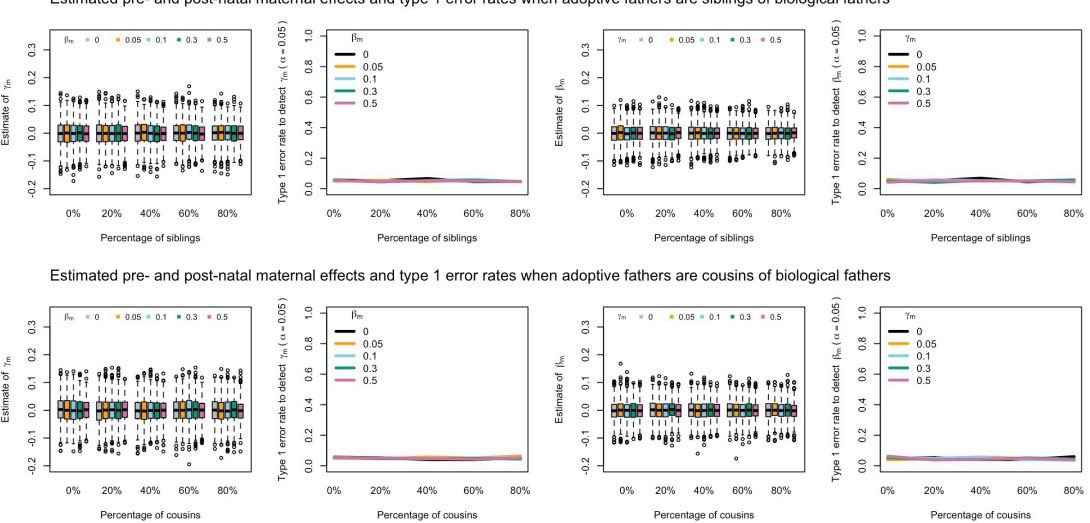

**Appendix 1—figure 9.** Estimated prenatal ($\gamma_m$) and postnatal ($\beta_m$) maternal effects and type 1 error rates whilst varying the percentage of adoptive fathers being siblings (top) or cousins (bottom) of biological fathers. An SEM which modelled this relationship correctly was fit to the data. Power was calculated using simulated data with 1000 biological trios, 4000 biological mother-offspring pairs, 1800 biological father-offspring pairs, 300,000 singletons, and 6000 adopted individuals. Prenatal maternal genetic effects were fixed to zero whilst the size of postnatal maternal genetic effects were varied (left panels); postnatal maternal genetic effects were fixed to zero whilst the size of prenatal maternal genetic effects were varied (right panels). Paternal effects ($\beta_p$) and offspring effects ($\beta_o$) were fixed to 0.1. The covariance between maternal and paternal genotypes was fixed to zero.

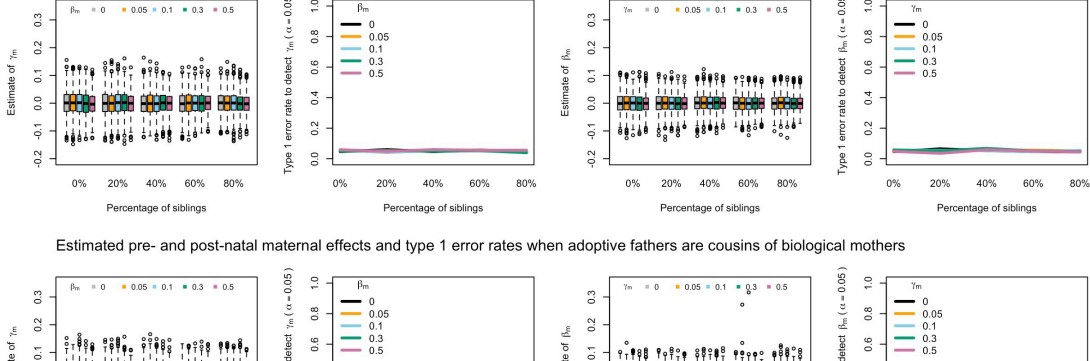

**Appendix 1—figure 10.** Estimated prenatal ($\gamma_m$) and postnatal ($\beta_m$) maternal effects and type 1 error rates whilst varying the percentage of adoptive fathers being siblings (top) or cousins (bottom) of biological mothers. An SEM which modelled this relationship correctly was fit to the data. Power was calculated using simulated data with 1000 biological trios, 4000 biological mother-offspring pairs, 1800 biological father-offspring pairs, 300,000 singletons, and 6000 adopted individuals. Prenatal maternal genetic effects were fixed to zero whilst the size of postnatal maternal genetic effects were varied (left panels); postnatal maternal genetic effects were fixed to zero whilst the size of prenatal maternal genetic effects were varied (right panels). Paternal effects ($\beta_p$) and offspring effects ($\beta_o$) were fixed to 0.1. The covariance between maternal and paternal genotypes was fixed to zero.

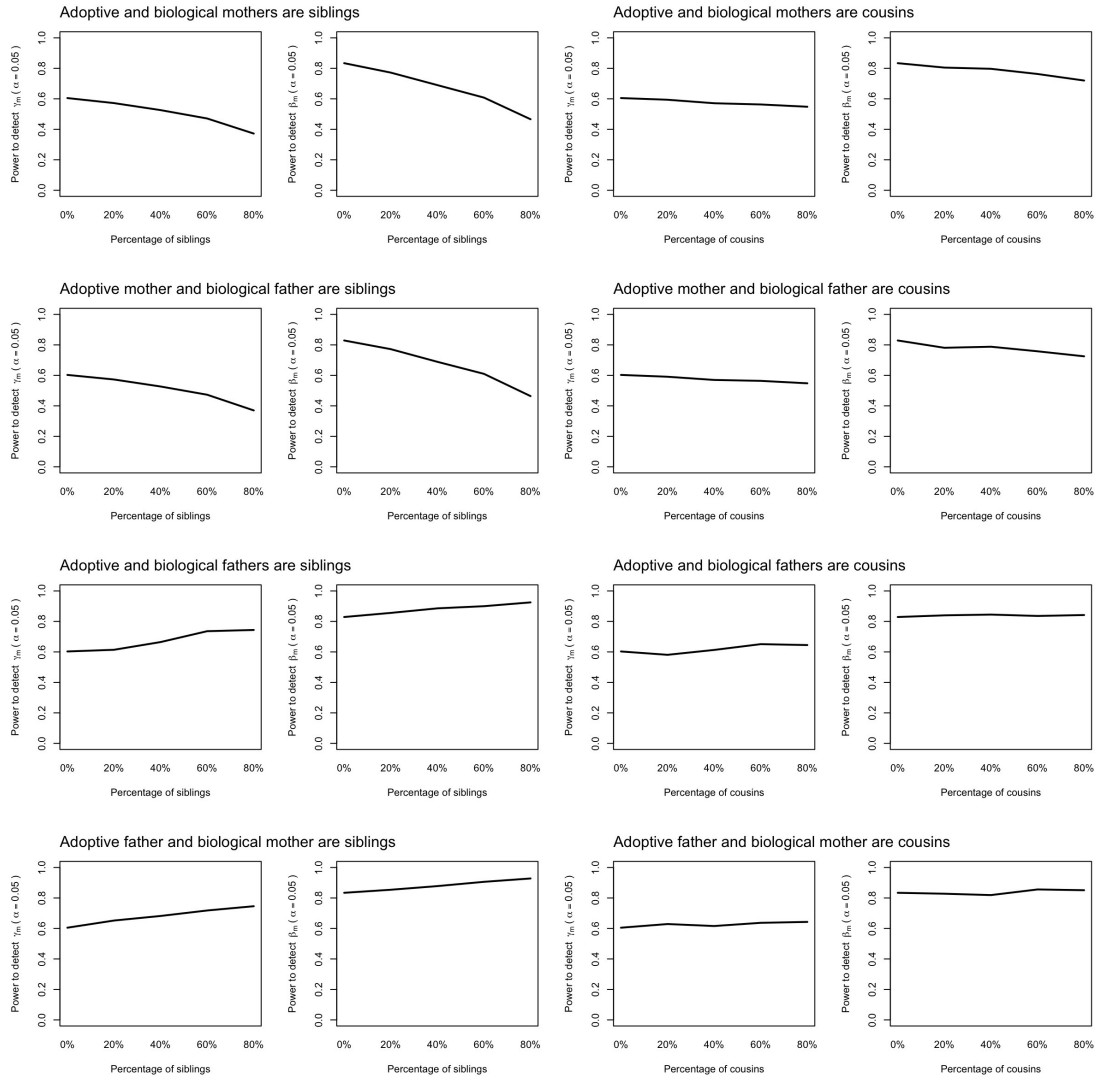

**Appendix 1—figure 11.** Power to detect prenatal ($\gamma_m$) and postnatal maternal ($\beta_m$) genetic effects when the relationship between adoptive and biological parents are correctly specified in the model. Power was calculated using simulated data assuming 1000 biological parent-offspring trios, 4000 biological mother-offspring pairs, 1800 biological father-offspring pairs, 300,000 singletons from non-adopted families, and 6000 adopted individuals. Prenatal maternal genetic effects, postnatal maternal genetic effects, paternal genetic effects ($\beta_p$) and offspring genetic effects ($\beta_o$) were fixed to 0.1. The covariance between maternal and paternal genotypes was fixed to 0. The percentage of adopted individuals whose adoptive parents were genetically related to their biological parents was varied (x-axis). Note that 0% corresponds to the situation where biological and adoptive parents are unrelated.

