## [Editor Report]

The authors propose an innovative and sound method to leverage the adoption of a design for disentangling prenatal and postnatal genetic effects. This work will be of interest to scientists interested in the intergenerational transmission of phenotypes through genetic pathways.

---

## [Decision Letter]

**Decision letter after peer review:**

Thank you for submitting your article "Using adopted individuals to partition indirect maternal genetic effects into prenatal and postnatal effects on offspring phenotypes" for consideration by *eLife*. Your article has been reviewed by 2 peer reviewers, and the evaluation has been overseen by George Perry as the Senior Editor. The following individual involved in review of your submission has agreed to reveal their identity: Stephane Paquin (Reviewer #2).

The Reviewing Editor has drafted this to help you prepare a revised submission.

The reviewers of your paper were positive about your work, but also identified several limitations that are not properly accounted in the current version of the paper. Alongside the critiques the reviews are so constructive that reasonable paths forward on each of the points raised below – primarily with thorough textual changes only – should be readily apparent.

*Reviewer #1 (Recommendations for the authors):*

GWAS data used for polygenic scores

The decision to include only genome-wide significant SNPs was not justified. Please could the authors fill the readers in on this? Especially for EA, a highly polygenic trait, why not include all SNPs? It is also not clear why they performed the sensitivity analysis with the less powered Okbay GWAS – perhaps this could be put in the Supplementary Information. Can the authors comment on how their results are affected by pre- and post-natal IGEs being captured by the original GWAS used to construct polygenic scores (discussed in Fletcher et al., 2021)?

Positioning among prior work

The authors could spell out more for readers their answer to the 'big picture' question of why it is useful to distinguish pre- and post-natal maternal effects, instead of just saying it's a natural question (see Introduction). Also in the introduction, it does not seem accurate to say that partitioning variation could yield mechanistic insights. Although the current contribution is very valuable, don't we need different kinds of designs and measures to understand mechanisms? In this regard, the authors should discuss the merits of their study compared to that of Armstrong-Carter et al. That study does not offer a principled way to separate pre- and post-natal effects, but it does try to work out which kinds of maternal phenotypes are involved in so-called pre-natal genetic nurture effects captured by EA PGS.

The authors should also engage more with other studies to clarify the novel aspects of their contribution. They describe in the Discussion the Domingue and Fletcher (2020) paper, but more relevant is Cheesman et al's., (2020) study design using the nonadopted-adopted polygenic score comparison in UKBB to estimate post-natal parental indirect genetic effects. Demange et al., (2021) estimated pre-natal parental indirect genetic effects by comparing estimates from adoption and sibling designs (sib-based estimates contain pre- and post-natal effects, but adoption only contain the latter). Although this was for a non-cognitive skills polygenic score and not EA or BW, it is worth discussing their approach and comparing against their estimates. The Demange et al., approach means that estimates of pre-natal maternal genetic effects may also include assortative mating and population stratification. There is a general paragraph on AM in the Discussion, but how do the authors think that AM and popstrat will load in their model? Recent evidence from UKBB suggests that it is important to consider the role of population stratification particularly when studying EA (Young et al., 2020).

Misc.

It is neat to have a negative control phenotype for post-natal genetic effects (birthweight). Can the authors try out a negative control for pre-natal genetic effects too? If the maternal pre-natal effect were significant for something paternal substance use, this would bolster the idea that relatedness among adoptees and their adoptive parents causes inflation of this estimate.

The use of breastfeeding data to identify those who were not adopted by relatives is smart. It would be worth checking how selected this sample is for some key traits (birthweight, EA). Also, could more information be leveraged to increase accuracy? Many participants just don't know the answer to that question, could have remembered incorrectly, and wetnurses (i.e., breastfeeding by non-relatives) also used to be common. Adoptions in later cohorts might have become more 'organized' (happening earlier in life and involving non-relatives). Perhaps it's worth cutting out some of the older adoptees? The authors could also try limiting the sample further to those who also report their birthweight.

Was a relatedness cut-off used to ensure that each family was independent?

The authors could cite Wang et al., (2021) when saying that it is unfeasible for paternal genetic effects to be so much bigger than maternal effects – this meta-analysis found equal effects of paternal and maternal EA PGS on offspring educational outcomes.

References

Armstrong-Carter E, Trejo S, Hill LJ, Crossley KL, Mason D, Domingue BW. The earliest origins of genetic nurture: The prenatal environment mediates the association between maternal genetics and child development. Psychological science. 2020 Jul;31(7):781-91.

Cheesman R, Hunjan A, Coleman JR, Ahmadzadeh Y, Plomin R, McAdams TA, Eley TC, Breen G. Comparison of adopted and nonadopted individuals reveals gene-environment interplay for education in the UK Biobank. Psychological science. 2020 May;31(5):582-91.

Demange PA, Hottenga JJ, Abdellaoui A, Eilertsen EM, Malanchini M, Domingue BW, de Zeeuw EL, Rimfeld K, Eley TC, Boomsma DI, van Bergen E. Estimating effects of parents' cognitive and non-cognitive skills on offspring education using polygenic scores. bioRxiv. 2021 Jan 1:2020-09.

Fletcher J, Wu Y, Li T, Lu Q. Interpreting polygenic score effects in sibling analysis. bioRxiv. 2021 Jan 1.

Wang B, Baldwin JR, Schoeler T, Cheesman R, Barkhuizen W, Dudbridge F, Bann D, Morris TT, Pingault JB. Robust genetic nurture effects on education: A systematic review and meta-analysis based on 38,654 families across 8 cohorts. The American Journal of Human Genetics. 2021 Aug 19.

Young AI, Nehzati SM, Lee C, Benonisdottir S, Cesarini D, Benjamin DJ, Turley P, Kong A. Mendelian imputation of parental genotypes for genome-wide estimation of direct and indirect genetic effects. BioRxiv. 2020 Jan 1.

*Reviewer #2 (Recommendations for the authors):*

I have no other specific recommendations. I think the approach and code provided are sound. I would mostly appreciate that the authors consider adding more details on the limitations (and possible extensions) of their approach regarding gene-environment interplay (both gene-environment moderation and gene-environment correlation).

---

## [Author Response]

Reviewer #1 (Recommendations for the authors):GWAS data used for polygenic scoresThe decision to include only genome-wide significant SNPs was not justified. Please could the authors fill the readers in on this? Especially for EA, a highly polygenic trait, why not include all SNPs? It is also not clear why they performed the sensitivity analysis with the less powered Okbay GWAS – perhaps this could be put in the Supplementary Information. Can the authors comment on how their results are affected by pre- and post-natal IGEs being captured by the original GWAS used to construct polygenic scores (discussed in Fletcher et al., 2021)?

We included only genome-wide significant SNPs in the construction of the EA score because the most recent GWAS meta-analysis by Lee et al., (2018) included UKBiobank individuals (and likewise for birthweight). We were concerned that sample overlap may inflate estimates of the association between the PRS and educational attainment (and any bias would be most severe for the hundreds of thousands of SNPs that did not reach genome-wide significance) which may in turn bias parameter estimates from our model. This was also our reason for including the less powered Okbay et al., GWAS as a sensitivity analysis. We have modified the text in the methods to make this clear for the reader:

“We included only genome-wide significant SNPs in the construction of these scores because the most recent GWAS meta-analysis of educational attainment by Lee et al., (2018) included UKBiobank individuals. We were concerned that sample overlap would inflate estimates of the association between the PRS and educational attainment (and any inflation would be most severe for the hundreds of thousands of SNPs that did not reach genome-wide significance) and in turn bias parameter estimates from our model. For this reason, we also performed a sensitivity analysis using a second set of PRS of educational attainment using 72 SNPs from a GWAS of non-UK Biobank individuals (Okbay et al., 2016).”

As suggested by the reviewer, we have moved the results of the Okbay et al., PRS to the Supplementary Information (Supplementary File 5).

The Fletcher et al., (2021) study (which it is important to note at this stage is only a pre-print and has not been peer-reviewed) involves using *weighted* polygenic risk scores in a sibling between/within design. In contrast, our design doesn’t use siblings, and we have also conducted analyses using both weighted *and unweighted* polygenic risk scores. Unweighted scores should not be influenced by the sort of biases that Fletcher et al., describe.

Positioning among prior workThe authors could spell out more for readers their answer to the 'big picture' question of why it is useful to distinguish pre- and post-natal maternal effects, instead of just saying it's a natural question (see Introduction). Also in the introduction, it does not seem accurate to say that partitioning variation could yield mechanistic insights. Although the current contribution is very valuable, don't we need different kinds of designs and measures to understand mechanisms? In this regard, the authors should discuss the merits of their study compared to that of Armstrong-Carter et al. That study does not offer a principled way to separate pre- and post-natal effects, but it does try to work out which kinds of maternal phenotypes are involved in so-called pre-natal genetic nurture effects captured by EA PGS.

We have changed the last two sentences of the first paragraph of the introduction to read:

“Given the increasing number of variants identified in GWAS that exhibit robust maternal genetic effects, a natural question to ask is whether these loci exert their effects on offspring phenotypes through intrauterine mechanisms, the postnatal environment, or both. Indeed, resolving maternal effects into prenatal and postnatal sources of variation could be a valuable first step in eventually elucidating the underlying mechanisms behind these associations (Armstrong-Carter et al., 2020), directing investigators to where they should focus their attention, and in the case of disease-related phenotypes, yielding potentially important information regarding the optimal timing of interventions. For example, the demonstration of maternal prenatal effects on offspring IQ/educational attainment, suggests that if the mediating factors that were responsible could be identified, then improvements in the prenatal care of mothers and their unborn babies which target these factors, could yield useful increases in offspring IQ/educational attainment.”

We have also added the following sentences to the discussion:

“Whilst our method does not provide explicit mechanistic insight into the reason for the existence of maternal genetic effects (merely a simple partitioning into pre- and postnatal components, and estimation as to the relative importance of each), follow up studies could help elucidate these mechanisms by e.g. examining the association between maternal PRS and different pre- and postnatal maternal phenotypes, and performing mediation and Mendelian randomization analyses (Armstrong-Carter et al., 2020; Evans et al., 2019).”

The authors should also engage more with other studies to clarify the novel aspects of their contribution. They describe in the Discussion the Domingue and Fletcher (2020) paper, but more relevant is Cheesman et al.,’s (2020) study design using the nonadopted-adopted polygenic score comparison in UKBB to estimate post-natal parental indirect genetic effects. Demange et al., (2021) estimated pre-natal parental indirect genetic effects by comparing estimates from adoption and sibling designs (sib-based estimates contain pre- and post-natal effects, but adoption only contain the latter). Although this was for a non-cognitive skills polygenic score and not EA or BW, it is worth discussing their approach and comparing against their estimates. The Demange et al., approach means that estimates of pre-natal maternal genetic effects may also include assortative mating and population stratification. There is a general paragraph on AM in the Discussion, but how do the authors think that AM and popstrat will load in their model? Recent evidence from UKBB suggests that it is important to consider the role of population stratification particularly when studying EA (Young et al., 2020).

In order to discuss the Cheesman et al., (2020) and Demange et al., (2021) results in relation to our findings, we have added the following text to the Discussion section:

“We are also not the first to have used adopted individuals in the UK Biobank to estimate the contribution of indirect genetic effects to phenotypic variability. Cheesman et al., (2020) contrasted adopted and non-adopted individuals in the UK Biobank to investigate evidence for indirect genetic effects on educational attainment and the existence of gene-environment correlation. The authors found that a genome-wide PRS for educational attainment was more strongly associated with educational attainment in non-adopted compared to adopted individuals consistent with the existence of indirect genetic effects on educational attainment. Likewise, G-REML estimates of SNP heritability were also lower in adopted individuals compared to non-adopted individuals (potentially due to the absence/lower amount of passive gene-environment correlation in adopted individuals).

In a previous study of cognitive and non-cognitive effects on educational attainment, Demange et al., (2021) used three different sorts of study design to estimate the contribution of indirect genetic effects on educational attainment. They estimated indirect genetic effects by comparing between family and within family genetic effects in sibling pairs (Fulker et al., 1999), by estimating the effect of non-transmitted parental alleles on offspring phenotype in parent-offspring trios, and by computing the difference in PRS associations between adopted and non-adopted individuals. In general, the authors found that estimates of indirect effects using the adoption study design were lower than those produced by the other two models. They interpreted this as being a consequence of the adoption study methodology not capturing the impact of prenatal maternal genetic effects (and also potentially being more robust to assortative mating and population stratification) in contrast to the two other study designs. The present study explicitly capitalizes on differences between adopted and non-adopted individuals in their pattern of genotype-trait covariances to estimate the relative size of pre- and postnatal maternal genetic effects and to provide formal statistical tests for their presence. Indeed, both Demange et al.,’s (2021) and our findings are consistent with the existence of prenatal effects on educational attainment (although we believe that at least part of this finding is an artefact and due to unmodelled biological relationships between some adopted children and their adoptive parents in the UK Biobank).”

In order to discuss the potential impact of population stratification and assortment on our model, we ran some basic simulations to gain intuition for their likely effect on our model (not shown), and have included the following text in the discussion:

“We have also not modelled the complex effects of assortative mating in our SEM other than including covariance terms between maternal and paternal genotypes and assuming equal genetic variances in parents and their offspring under random mating (i.e. this is equivalent to assuming one round of assortative mating in the parental generation). Positive assortment induces a number of complications when attempting to decompose the offspring phenotypic variance into its constituent sources of variation, including increasing homozygosity and the genetic variance at loci for the trait undergoing assortment relative to that expected under Hardy-Weinberg equilibrium, and inducing correlations between assorting loci across the rest of the genome both within and between individuals in the same family. Spousal correlations for phenotypes (educational attainment: r = 0.285; birthweight: r = 0.076) and PRS (educational attainment: r = 0.117; birthweight: r = 0.068) in complete parent-offspring trios in the UK Biobank suggest that assortative mating is likely present for educational attainment, and potentially to a lesser degree for (phenotypes correlated with) birthweight. Recent work by Balboa et al., (2021) and Kim et al., (2021) have shown how PRS of transmitted and non-transmitted alleles in parent-offspring trios can be used to estimate direct and indirect genetic effects on offspring phenotype and the variation attributable to the environmental influence of parents on offspring under phenotypic assortment ^2,6^. It is possible that our basic model could be extended in a similar fashion to incorporate the effect of assortment and estimate some of these effects also.

Also relevant to this discussion is a series of data simulations performed by Demange et al., (2021) who found that estimates of indirect genetic effects obtained in adoption designs (i.e. which estimate the size of indirect genetic effects by taking the difference between a PRS-trait association in adopted individuals and a PRS-trait association in non-adopted individuals) were less biased by assortative mating and population stratification than estimates of indirect genetic effects obtained in biologically related parent-offspring trio designs (i.e. where estimates of indirect genetic effects are obtained by regressing offspring phenotype on parental genotype conditional on offspring genotype). In the case of population stratification, the intuition behind this observation is that the contaminating effects of substructure are present in both adopted and non-adopted individuals. This contamination cancels out when the PRS-trait association in adopted individuals is subtracted from the PRS-trait association in non-adopted individuals. However, this is not the case in parent-offspring trios, where estimates of indirect genetic effects remain contaminated.

In the context of our SEM, we expect that unmodelled population stratification will bias estimates of maternal prenatal and postnatal effects away from the null since population stratification will increase the covariance between maternal genotype and offspring phenotype, as well as the covariance between offspring genotype and phenotype in adopted and non-adopted individuals. Because we have limited our empirical analyses to individuals of white European ancestry and residualized the phenotypes for the first five genetic principal components prior to analysis in the SEM, any influence due to population stratification is likely to have been minor in our empirical analyses in the UK Biobank. The influence of assortative mating on our model is more difficult to predict, but our limited exploration of this issue by simulation (data not shown) suggests that the effect of assortative mating on parameter estimates from our SEM is likely to be minor.”

Misc.It is neat to have a negative control phenotype for post-natal genetic effects (birthweight). Can the authors try out a negative control for pre-natal genetic effects too? If the maternal pre-natal effect were significant for something paternal substance use, this would bolster the idea that relatedness among adoptees and their adoptive parents causes inflation of this estimate.

We like the reviewer’s idea of investigating a control phenotype, but it is exceedingly difficult to think of a trait for which (a) there is confirmed evidence for maternal genetic effects, (b) the maternal genetic effects are so strong that we have adequate power to demonstrate and partition them in our sample, and (c) we can be confident that the maternal genetic effects are postnatal but not prenatal effects.

That being said, we have hit upon the idea of using individual’s report of maternal hypertension as a control phenotype. In theory, an individual’s report of maternal hypertension should show (unconditional) association with a maternal PRS for hypertension in biological families (i.e. G1, G2 in our model) and with offspring PRS (i.e. through the latent biological maternal genotype as in G3, G4 in our model). In contrast, in adoptive families (G5-G7), and assuming adoptive parents are not biologically related to their adopted children, we would expect no association between reports of maternal hypertension (in adoptive mothers) and their adopted offspring’s PRS. Under these conditions, our SEM should “correctly” partition the maternal genetic effect entirely into a postnatal maternal genetic component. In contrast, if adopted individuals are biologically related to their adoptive mothers, then this will tend to lead to spurious associations between adopted individuals’ PRS and (adopted mothers’) maternal hypertension. Within the context of our SEM, this will increase evidence for a prenatal maternal genetic effect and decrease evidence for a postnatal maternal genetic effect. In other words, the degree to which our SEM produces evidence for a prenatal maternal genetic effect for individual’s report of maternal hypertension provides evidence consistent with the possibility that at least some adoptive individuals in the UK Biobank are biologically related to their adoptive parents.

The results of our analyses are as follows, see Author response table 1.

**Author response table 1. sa2table1:** 

Effect	matrix	Estimate	Std.Error	P-value
GRS	V	25.317	0.190	0.000
Fetal effect	B_FY	0.000	0.002	0.997
Postnatal maternal effect	B_MY	0.007	0.004	0.113
Paternal effect	B_PY	-0.001	0.003	0.697
Prenatal maternal effect	G_MY	0.003	0.005	0.586
e1	E1	0.206	0.001	0.000
e2	E2	0.179	0.005	0.000
rho	R	-0.045	0.369	0.907

In summary, estimates for the fetal and paternal genetic effects are what we would expect (i.e. close to zero). In contrast, the coefficients for the maternal genetic effects are larger and in the case of the pre-natal maternal genetic effect, potentially consistent with at least a proportion of adopted individuals in the UK Biobank being biologically related to their adoptive parents. However, a complication of this analysis is that self-reported maternal hypertension is a dichotomous, rather than quantitative phenotype. The corollary is that whilst we expect the point estimates of the parameters to be consistent estimates of the true population values in a large sample such as the UK Biobank (on the observed scale), the standard errors and tests of significance will not be appropriate for this phenotype. We have spoken extensively with the developers of OpenMx (Mike Neale, Joshua Pritiken) about the possibility of fitting the SEM as an ordinal model, but the developers have told us that this is not computationally feasible at the present time. For this reason, our preference is not to include these analyses in the published manuscript, but we have included them here for the reviewers’ information and in support of our contention that there is likely to be some degree of relatedness between adopted individuals and their adoptive parents in the UK Biobank.

*The use of breastfeeding data to identify those who were not adopted by relatives is smart. It would be worth checking how selected this sample is for some key traits (birthweight, EA). Also, could more information be leveraged to increase accuracy? Many participants just don’t know the answer to that question, could have remembered incorrectly, and wetnurses (i.e., breastfeeding by non-relatives) also used to be common. Adoptions in later cohorts might have become more ‘organized’ (happening earlier in life and involving non-relatives). Perhaps it’s worth cutting out some of the older adoptees? The authors could also try limiting the sample further to those who also report their birthweight.*

Adopted individuals who knew their birthweight were more educated on average than those who did not, but did not differ on average in their birthweight.

**Author response table 2. sa2table2:** 

	Adopted individuals who knew their breastfeeding status	Adopted individuals who did not know their breastfeeding status
Years of education (mean (SD))	13.97 (4.98)	13.15 (5.00)
Birthweight (kg; mean (SD))	3.30 (0.43)	3.34 (0.42)

Following the reviewer’s suggestion, we stratified adopted individuals into those above and below age 60 and re-ran the analysis for EA. The results suggests that the deflation in post-natal maternal effects comes from adoptions in later cohorts, see Author response table 3:

**Author response table 3. sa2table3:** 

Effect	matrix	Adopted individuals with age<60 removed (N_G5=2756)	Adopted individuals with age>=60 removed (N_G5=2422)
Estimate	Std.Error	P-value	Estimate	Std.Error	P-value
GRS	V	571.885	4.747	0.000	571.943	5.006	0.000
Fetal effect	B_FY	0.026	0.005	0.000	0.025	0.005	0.000
Postnatal maternal effect	B_MY	-0.019	0.009	0.046	0.007	0.0100	0.482
Paternal effect	B_PY	0.018	0.007	0.008	0.019	0.007	0.004
Prenatal maternal effect	G_MY	0.039	0.011	0.001	0.014	0.012	0.235
e1	E1	23.897	0.119	0.000	23.870	0.126	0.000
e2	E2	21.544	0.857	0.000	24.335	0.727	0.000
rho	R	58.208	9.397	0.000	58.262	9.956	0.000

It is possible that more recent adoptions are more likely to involve biological relatives of the adopted individual. We also removed individuals who reported their birthweight in addition to breastfeeding data. However, this only removed a further 223 adopted individuals, and did not appreciably affect our results. Given that the analyses have not materially affected our interpretation of the results, we have elected not to include them in the main manuscript.

Was a relatedness cut-off used to ensure that each family was independent?

We would like to thank the reviewer for bringing this to our attention. The relatedness cut-off was set at 3^rd^ degree of relatives defined by the pair-wise kinship >0.044 (Manichaikul et al., 2010). In our original analysis, families in each of the 7 groups were not related based on this cut-off. However, we did not remove families that were related across groups. We have now removed all related families from our analysis. We maximized the number of families from adopted singletons (G5), followed by trios (G1), mother-offspring duos (G2), and then father-offspring duos (G3). Results from the new analysis are similar to our original analysis and do not affect our conclusions.

The authors could cite Wang et al., (2021) when saying that it is unfeasible for paternal genetic effects to be so much bigger than maternal effects – this meta-analysis found equal effects of paternal and maternal EA PGS on offspring educational outcomes.

We thank the reviewer for this suggestion- however, our analyses actually showed *similar* effect sizes for maternal and paternal genetic effects (i.e. bearing in mind that total maternal effects = prenatal maternal effects + post-natal maternal effects). Nevertheless, we appreciate that equal paternal and maternal effects on offspring educational attainment implies the absence of maternal prenatal effects. We have therefore at the reviewer’s suggestion included the Wang et al., reference and modified some of the text in the discussion:

“The absence of a significant postnatal maternal genetic effect on offspring educational attainment was surprising given that maternal genetic effects should be mediated through maternal educational attainment, and therefore should mostly involve postnatal pathways (although it is possible that some of the relationship between maternal genotype and offspring educational attainment could be mediated through prenatal effects- e.g. less educated mothers consuming more alcohol during pregnancy which then has adverse effects on offspring cognitive development and educational attainment etc), and previous studies have shown that the effect of paternal and maternal PRS on educational attainment are roughly similar (Wang et al., 2021) i.e. suggesting the absence of prenatal maternal genetic effects.”

Reviewer #2 (Recommendations for the authors):I have no other specific recommendations. I think the approach and code provided are sound. I would mostly appreciate that the authors consider adding more details on the limitations (and possible extensions) of their approach regarding gene-environment interplay (both gene-environment moderation and gene-environment correlation).

We have added this information to Supplementary File 6 which discusses gene-environment interaction, gene-environment correlation and other limitations.